# Self-assembly of CIP4 drives actin-mediated asymmetric pit-closing in clathrin-mediated endocytosis

Yiming Yu[1] & Shige H. Yoshimura [1]✉

Clathrin-mediated endocytosis is pivotal to signal transduction pathways between the extracellular environment and the intracellular space. Evidence from live-cell imaging and super-resolution microscopy of mammalian cells suggests an asymmetric distribution of actin fibres near the clathrin-coated pit, which induces asymmetric pit-closing rather than radial constriction. However, detailed molecular mechanisms of this 'asymmetricity' remain elusive. Herein, we used high-speed atomic force microscopy to demonstrate that CIP4, a multi-domain protein with a classic F-BAR domain and intrinsically disordered regions, is necessary for asymmetric pit-closing. Strong self-assembly of CIP4 via intrinsically disordered regions, together with stereo-specific interactions with the curved membrane and actin-regulating proteins, generates a small actin-rich environment near the pit, which deforms the membrane and closes the pit. Our results provide mechanistic insights into how disordered and structured domain collaboration promotes spatio-temporal actin polymerisation near the plasma membrane.

Clathrin-mediated endocytosis (CME) is necessary for several cellular processes, including low-density lipoprotein uptake, epidermal growth factor receptor signalling transduction, synaptic vesicle formation, and cell migration[1–3]. Clathrin-dependent endocytic machinery is highly disciplined in recruiting and disassembling over 60 endocytic proteins in an ordered sequence[4]. Clathrin-coated pit (CCP) initiation requires the assembly of several adaptors and scaffold proteins, including the Fes/Cip4 homology Bin/amphiphysin/Rvs (F-BAR) domain-only protein, Eps15, AP2 complex, and intersectin[5–10]. Polymerised clathrin forms a hexagonal lattice that curves the plasma membrane and gradually drives CCP invagination[11]. In the late CME stage, highly invaginated CCPs are detached from the membrane via dynamin-mediated constriction and scission and then transported to the cytosol for further processing[12,13].

The pit-closing step of CME has recently attracted much research attention owing to the series of processes involved, including membrane deformation, fusion, and scission. In addition to a canonical dynamin-dependent constriction mechanism, actin is crucial to the endocytic progression in yeast and mammals[14]. In yeast cells, actin assembles symmetrically at the base of the CCP in a radial distribution and constricts the membrane tubule[15]. In mammalian cells, actin functions as a force generator that counteracts membrane tension, elongates the membrane tubules, and assists in forming deeply-invaginated vesicles[16–18]. Furthermore, actin coordinates other endocytic proteins, such as dynamin and BAR domain proteins, to promote the constriction and scission of the membrane tubules before vesicle translocation[19–21].

Asymmetric distribution revealed via total internal reflection fluorescence microscopy is an interesting characteristic of the actin cytoskeleton around CCP[22]. Recent progress in super-resolution techniques further revealed that many endocytic proteins possess unique localisation around the clathrin lattice[23] and that actin and some associated proteins, including the neural Wiskott–Aldrich syndrome protein (N-WASP) and actin-related proteins-2/3 (Arp2/3) complex, asymmetrically assemble near the CCP[24]. Correspondingly, live-cell imaging with ion conductance microscopy revealed a membrane bulge that asymmetrically formed around the CCP and gradually covered that area[25].

Correlative imaging of high-speed atomic force microscopy (HS-AFM) and confocal laser-scanning microscopy (CLSM) is powerful for

[1]Graduate School of Biostudies, Kyoto University, Kyoto 606-8501, Japan. ✉e-mail: yoshimura@lif.kyoto-u.ac.jp

investigating the morphological changes of the plasma membrane during endocytosis[26]. Our previous study using this technique has revealed a good correlation between the progress of membrane invagination and the clathrin signal accumulation supporting the hotly debated 'constant curvature model'[27,28]. Moreover, we observed an asymmetric closing process dependent on Arp2/3-mediated actin polymerisation[27]. A lateral constriction is energetically more favourable for membrane fusion than a radial constriction[29]. Therefore, the actin-dependent asymmetric closing mechanism seems crucial in CME; however, the detailed molecular mechanism generating the asymmetric assembly of actin machinery remains elusive.

Herein, we studied the molecular mechanism underlying asymmetric actin assembly and the consequent generation of the membrane bulge near the CCP. We used live-cell HS-AFM and fluorescence-based imaging to examine the involvement of F-BAR domain-containing proteins (CIP4, FBP17, and Syndapin 2), which carry structured domains to interact with actin regulatory proteins and long, disordered domains in CME. Our results provide mechanistic insights into how a multi-domain protein can spatio-temporally recruit and orchestrate the actin machinery near the CCP via phase separation of disordered domains and stereospecific interactions between structured domains.

## Results

### Asymmetric CCP closing

Using HS-AFM, CME was described as a group of near-circular invaginations with diameters of 150–400 nm and lifetimes of 40 to ~300 s[27]. This prior observation revealed various dynamic plasma membrane processes during the CME. Three different closing patterns, namely, 'asymmetric', 'symmetric', and 'undetermined', were identified (Fig. 1a, Supplementary Fig. 1a–c). The asymmetric pattern is the dominant closing pattern that corresponds to ~70% of total closing events (Fig. 1b), in which a membrane bulge emanates from one side (or a limited area) of the CCP and covers the pit at the end (Fig. 1a, Supplementary Fig. 1a, Supplementary Movie 1). The diameter of the pit plummeted within 10–20 s before complete closing (Fig. 1a, c, Supplementary Fig. 1a). In the symmetric pattern (~20% of total events, Fig. 1b), the pit slowly shrank isotropically and completely closed with a weak membrane bulge (Fig. 1a, c, Supplementary Fig. 1b, Supplementary Movie 2). A small portion of the pits (~10% of the total events, Fig. 1b) closed within 10–20 s but without detectable bulges (Fig. 1a, c, Supplementary Fig. 1c, Supplementary Movie 3). Moreover, scanning with a higher time resolution (0.5 frames/s) did not reveal a membrane bulge around these pits (Supplementary Fig. 1d); therefore, they were classified as 'undetermined'.

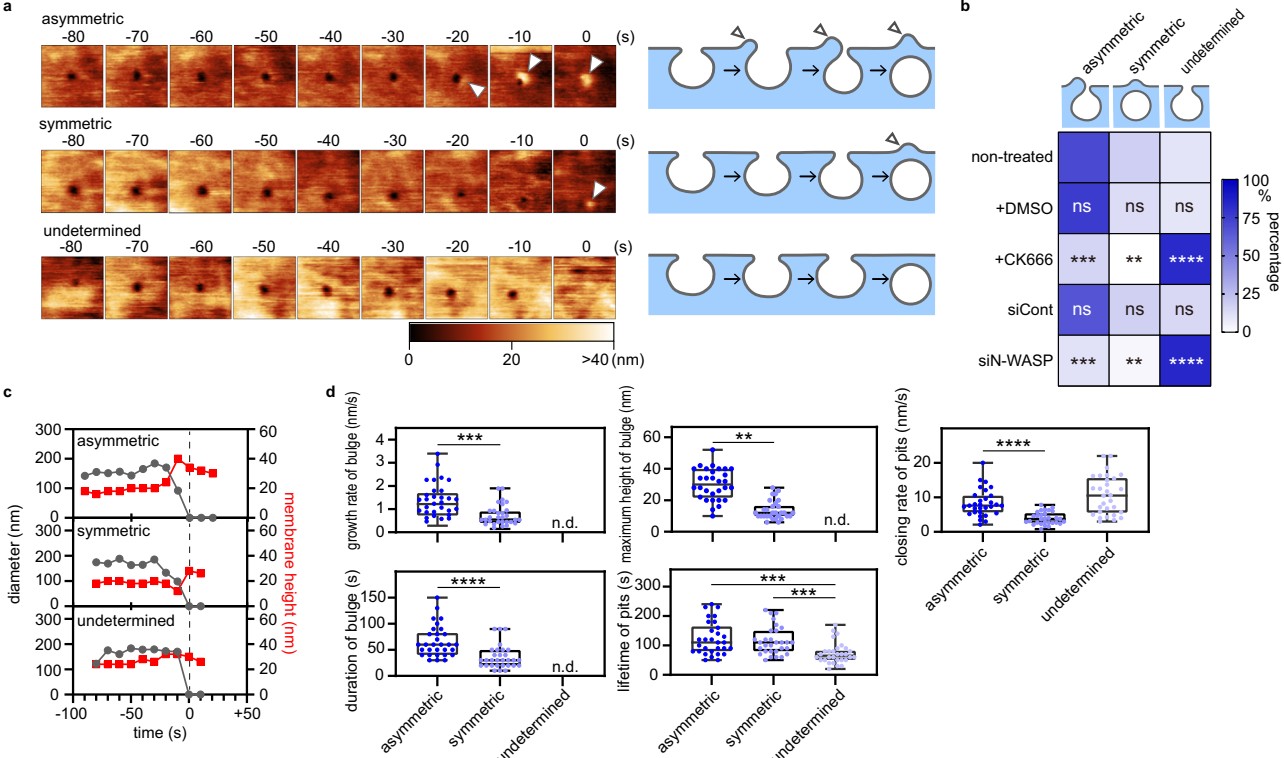

**Fig. 1 | Asymmetric closing pattern of the CCP. a** Time-lapse HS-AFM images and schematic illustrations of the asymmetric, symmetric, and undetermined closing patterns of the CCP identified in Cos7 cells. Images were taken every 10 s. Time 0 was when the pit completely closed on the HS-AFM image. Arrowheads indicate the membrane bulge, and the height information of the AFM image is presented using a colour bar. The image size is 1.0 × 1.0 μm². Representative images are presented here, and additional images are provided in Supplementary Fig. 1a–c. **b** Heat map of the ratio of the three CCP closing patterns in non-treated Cos7 cells, Cos7 cells treated with DMSO or CK666, and Cos7 cells transfected with siRNA for the control (luciferase) or N-WASP (N = 4 biologically independent cells, for each condition). P values were calculated by comparing the ratio of the same closing pattern between the non-treated and treated group using a two-tailed Student's t test and indicated in each block, **P < 0.01; ***P < 0.001; ****P < 0.0001; ns: not statistically significant, α = 0.05. The exact P values are provided in Source Data. **c** The diameter of CCP

(grey) and the maximum membrane height at the pit area (red) are plotted against time for the three different closing patterns. A representative case is illustrated in (**a**). Additional cases are shown in Supplementary Fig. 1a–c. Time 0 was when the pit completely closed and is indicated with a black dotted line. **d** The growth rate, maximum height, and duration of the bulge, and the total lifetime and the closing rate of CCP in individual closing patterns are plotted (N = 30 pits examined over three independent experiments, for each pattern). The median and lower and upper quartiles are indicated in the box. The minimum and maximum values are indicated with the upper and lower whiskers, respectively. In undetermined cases, the height, duration, and growth rate of the membrane bulge were not defined (n.d.). P values were calculated using the two-tailed Student's t test, **P < 0.01; ***P < 0.001; ****P < 0.0001, α = 0.05. The exact P values are provided in Source Data.

The asymmetric bulge grew faster (increase in the membrane height per unit time), higher (the maximum height of the bulge), and continued for a longer period (duration of the bulge) than did the symmetric bulge (Fig. 1d). However, CCPs with the asymmetric or symmetric closing patterns had a similar total lifetime (Fig. 1d). The closing rate (reduction in the diameter per unit time) also differed: the asymmetric pattern ($8.3 \pm 3.9$ nm/s) was faster than the symmetric one ($3.9 \pm 1.8$ nm/s) (Fig. 1d), suggesting that the pit-closure is accelerated more by the asymmetric bulge than the symmetric bulge. Treatment with CK666, an Arp2/3 complex inhibitor[30], and knockdown (KD) of N-WASP (Supplementary Fig. 1e), an upstream activator of Arp2/3[31], promoted the frequency of newly formed CCPs while reducing the frequency of asymmetric and symmetric bulges to ~20% and 0%, respectively (Fig. 1b, Supplementary Fig. 1f, g). This result demonstrates that N-WASP- and Arp2/3-dependent actin polymerisation is required for bulge formation.

## CIP4 is necessary for asymmetric bulge formation
We further identified the proteins that potentially mediate the asymmetric closing process. F-BAR domain-containing proteins were ideal candidates to coordinate actin reorganisation with membrane deformation during CME owing to their membrane deformation and actin-associated properties[32,33]. CIP4, Syndapin 2 (Synd2, also called Pacsin2), and FBP17 are classic F-BAR domain proteins localised at CCPs[22], and Cos7 cells express all these proteins (Supplementary Fig. 2a, b). The time-lapse fluorescence observations of Cos7 cells expressing mCherry-fused CIP4, Synd2, or FBP17 with an enhanced green fluorescent protein (EGFP)-fused clathrin light chain B (CLTB) revealed different timings and kinetics of their assembly at the CCP (Fig. 2a, Supplementary Fig. 3a–c). When the disappearance of the clathrin signal was set as time 0, CIP4, Synd2, and FBP17 appeared at the CCP at $-26.0 \pm 7.4$, $-16.0 \pm 5.5$, and $-10.5 \pm 4.0$ s and disassembled at $-5.0 \pm 2.3$, $-7.8 \pm 4.4$, and $-2.0 \pm 6.0$ s, respectively (Fig. 2b). CIP4 KD with two siRNA species (one for the coding region, siCIP4_coding; one for the 3' untranslated region, siCIP4_3'UTR) in Cos7 cells reduced the frequency of the asymmetric bulges (from ~70 to <30%) and increased the undetermined pattern (from ~10% to ~70%) (Fig. 2c, Supplementary Fig. 2c–e). Moreover, overexpression of EGFP-fused CIP4 in CIP4-KD (siCIP4_3'UTR) cells rescued the asymmetric bulge (Fig. 2c, Supplementary Fig. 2d). By contrast, Synd2 or FBP17 KD did not markedly affect any closing patterns (Fig. 2c, Supplementary Fig. 2c, d). Nevertheless, the overall frequency of CME was unaffected (Supplementary Fig. 2f). These results demonstrate that CIP4, but not FBP17 or Synd2, is necessary for the asymmetric bulge.

Correlative imaging of HS-AFM and CLSM revealed a clear correlation between the assembly of CIP4 and the bulge formation. When the disappearance of the clathrin signal was set as time 0, CIP4 was observed at $-33.9 \pm 6.5$ s under CLSM, and the membrane bulge formed at $-30.0 \pm 8.2$ s under the HS-AFM (Fig. 2d, Supplementary Fig. 2g), indicating a strong association between CIP4 and bulge formation.

The relative position of CIP4 and the clathrin signal was determined using time-lapse super-resolution structural illumination microscopy (SIM) to confirm the asymmetric localisation of CIP4. The CIP4 signal started to appear at the determined position offset from clathrin, with a centre-to-centre distance within the first 5 s of $126.8 \pm 66.6$ nm ($N = 160$) (Fig. 2e, f, Supplementary Fig. 2h). As the CIP4 signal increased, the centre-to-centre distance fluctuated with a decreasing tendency until both signals disappeared (Fig. 2e, Supplementary Fig. 2h). Thus, we concluded that CIP4 was asymmetrically assembled at the CME site.

## Flanking disordered region is vital for CIP4 assembly in CCP
The CIP4 domain required for self-assembly around CCP was subsequently determined. CIP4 and FBP17 belong to the same subfamily of F-BAR proteins and have conserved F-BAR and SH3 domains at the amino- and carboxyl-termini, respectively (Supplementary Fig. 3d, e). The middle region between these conserved domains is less conserved and mostly disordered without any known binding motifs for other endocytic proteins (Fig. 3a, Supplementary Fig. 3d, e). In CIP4 and FBP17, but not Synd2, the middle-disordered region is intervened by a conserved G protein-binding homology region 1 (HR1); therefore, it is named the flanking disordered region (FDR; Fig. 3a, Supplementary Fig. 3d).

The CIP4 assembly at the CCP was unaffected by SH3 domain deletion (CIP4(ΔSH3); Fig. 3a–c, Supplementary Fig. 4a). BAR domain and the FDR were insufficient for CIP4 assembly (Fig. 3b), indicating that the F-BAR domain and FDR are necessary for CIP4 to assemble at the CCP. Notably, the assembly of chimeric molecules carrying the Synd2-BAR and CIP4-FDR (SCC and SC chimaeras in Fig. 3a) to the CCP starts at a time point similar to that of full-length CIP4 (Fig. 3b, c, Supplementary Fig. 4a). By contrast, the CS mutant, which carries the CIP4-BAR and Synd2-IDR (Fig. 3a), did not assemble at the CCP (Fig. 3b). Therefore, these results demonstrate that the FDR of CIP4 is a major determinant of the self-assembly at the CCP. However, the BAR domain is also necessary.

## SH3 domain is required for membrane bulge formation
In contrast to the assembly at the CCP, the SH3 domain was required to form the membrane bulge. The SC chimaera expression, which could assemble at the CCP (Fig. 3b), failed to rescue the CIP4-KD phenotype; the frequency of the asymmetric membrane bulge was not recovered by the SC mutant expression (Fig. 3d). By contrast, the SCC chimaera, which also carries the SH3 domain of CIP4, could rescue the KD phenotype (Fig. 3d), indicating that the SH3 domain was necessary for forming the membrane bulge after the assembly at the CCP.

CIP4 binds and activates N-WASP via the SH3 domain[34]. The activated N-WASP interacts with Cdc42 through its GTPase binding domain (GBD) (Supplementary Fig. 4b, c) and recruits the Arp2/3 complex to promote actin polymerisation[35]. In our experimental system, the assembly of N-WASP at the CCP was also observed; it appeared at the CCP immediately after CIP4 (Supplementary Fig. 4d, e). Therefore, we examined whether the interaction between CIP4 and N-WASP is necessary for the asymmetric bulge. A point mutation in the SH3 domain of Toca1 (W518K) inhibited the interaction with N-WASP[36]. Mutating the corresponding residue in the CIP4-SH3 domain (W524K) also reduced the interaction with N-WASP (Fig. 3e, Supplementary Fig. 4f). In agreement with this finding, the SCC chimaera carrying the W524K mutation failed to rescue the CIP4-KD phenotype (Fig. 3d), indicating that the interaction with CIP4-SH3 and N-WASP is necessary for forming the asymmetric bulge.

## Cdc42 recruits CIP4 to the CCP site
Next, we examined FDR's involvement in the asymmetric assembly of CIP4 at the CCP. We first focused on the interaction with Cdc42[36]. A pull-down assay demonstrated that the CIP4-FDR had a higher affinity to the active (GTPγS-loaded) form of Cdc42 than to the FBP17-FDR and the IDR of Synd2 (Fig. 4a, Supplementary Fig. 5a). Substitutions of three amino acids within HR1 (MGD to IST, Supplementary Fig. 5b), which abolish the interaction with Cdc42 in Toca1-HR1[35], also attenuated the interaction with Cdc42 (Fig. 4a, Supplementary Fig. 5a). The assembly of full-length CIP4 carrying the MGD-IST mutation at the CCP started at $-12.5 \pm 5.2$ s (Fig. 4b, c, Supplementary Fig. 5c), which was delayed compared to that of the wild-type (WT), indicating that the activated Cdc42 is required for CIP4 assembly at the CCP.

The interaction with active Cdc42 was important for CIP4 assembly at the CCP and asymmetric bulge formation. The overexpression of a dominant-negative form of Cdc42(T17N), but not the WT, severely impaired the asymmetric bulge (to 25%) compared to that of the non-transfected cells (Fig. 4d). In addition, the SCC chimaera carrying the

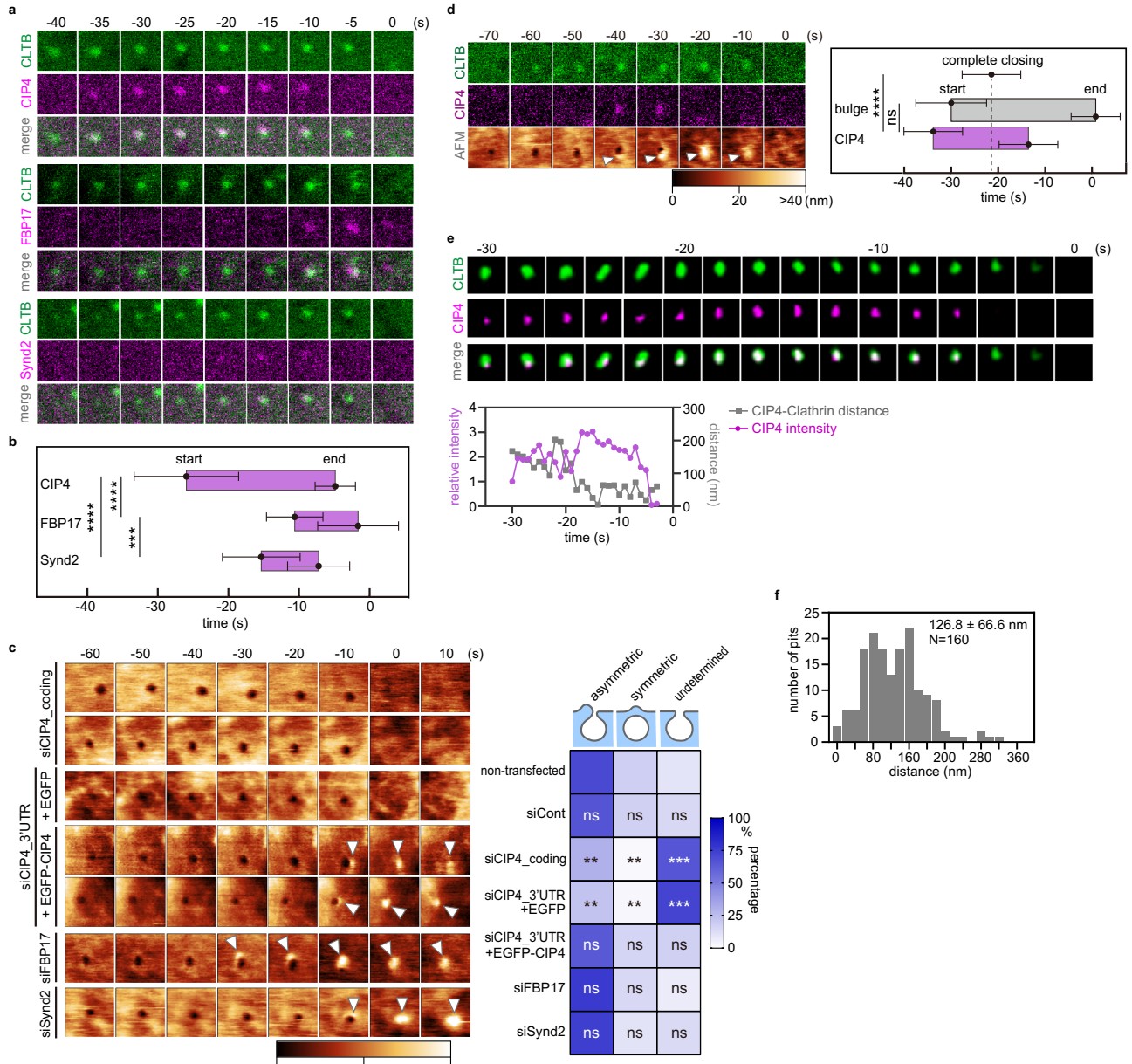

**Fig. 2 | CIP4 is necessary for asymmetric bulge formation. a** Time-lapse fluorescence images obtained from Cos7 cells expressing EGFP-fused CLTB and mCherry-fused CIP4, FBP17, or Synd2. Images of $1.0 \times 1.0\ \mu m^2$ were acquired. Time 0 was when the clathrin signal disappeared. Additional images are provided in Supplementary Fig. 3a–c. **b** The assembly profile of the F-BAR domain proteins. The times when the fluorescence signal appeared and disappeared at the CCP area are defined as 'start' and 'end', respectively, and are plotted ($N = 20$ pits examined over three independent experiments, for each condition). Time 0 was when the clathrin signal disappeared. *P* values were calculated using the two-tailed Student's *t* test, \*\*\*$P < 0.001$; \*\*\*\*$P < 0.0001$, $\alpha = 0.05$. The exact *P* values are provided in Source Data. Error bars represent the standard deviation. **c** (Left) HS-AFM images of Cos7 cells transfected with siRNA for CIP4 (siCIP4_coding, siCIP4_3'UTR), FBP17, or Synd2 followed by EGFP or EGFP-fused CIP4 overexpression. Additional cases are illustrated in Supplementary Fig. 2d. Time 0 was when the pit completely closed on the HS-AFM image. The arrowheads indicate the membrane bulge, and the height information of the AFM image is presented using a colour bar. Image size: $1.0 \times 1.0\ \mu m^2$. (Right) Heat map of the ratio of three closing patterns in each condition described in the left panel ($N = 4$ biologically independent cells, for each condition). *P* values were calculated by comparing the ratio of the same closing pattern between the non-transfected and transfected group using the two-tailed Student's *t* test and are indicated in each block, \*\*$P < 0.01$; \*\*\*$P < 0.001$; ns: not

statistically significant, $\alpha = 0.05$. The exact *P* values are provided in Source Data. **d** (Left) Correlative imaging of HS-AFM and CLSM. Cos7 cells expressing EGFP-fused CLTB and mCherry-fused CIP4 were imaged. Time 0 was when the pit completely closed on the HS-AFM image. The arrowheads indicate the membrane bulge, and the height information of the AFM image is presented using a colour bar. The image size is $1.0 \times 1.0\ \mu m^2$. Additional cases are presented in Supplementary Fig. 2g. (Right) The assembly profile of CIP4 and the membrane bulge. The time points when the membrane bulge and the CIP4 signal appeared and disappeared at the CCP area were defined as 'start' and 'end' points, respectively, and are plotted ($N = 16$ pits examined over three independent experiments). *P* values were calculated using the two-tailed Student's *t* test, \*\*\*\*$P < 0.0001$; ns: not statistically significant, $\alpha = 0.05$. The exact *P* values are provided in Source Data. Error bars represent the standard deviation. **e** Time-lapse super-resolution images of Cos7 cells expressing EGFP-fused CLTB and mCherry-fused CIP4. Images of $0.5 \times 0.5\ \mu m^2$ were taken every second and displayed every 2 s. The fluorescence intensity of CIP4 and the x–y distance between the centroids of clathrin and CIP are plotted against time (bottom). Time 0 was when the clathrin signal disappeared. Additional cases are illustrated in Supplementary Fig. 2h. **f** The histogram of the x–y distance between the centroids of the clathrin and CIP4 fluorescence spots within the first 5 s of CIP4 assembly ($N = 160$, from 32 CME events examined over three independent experiments).

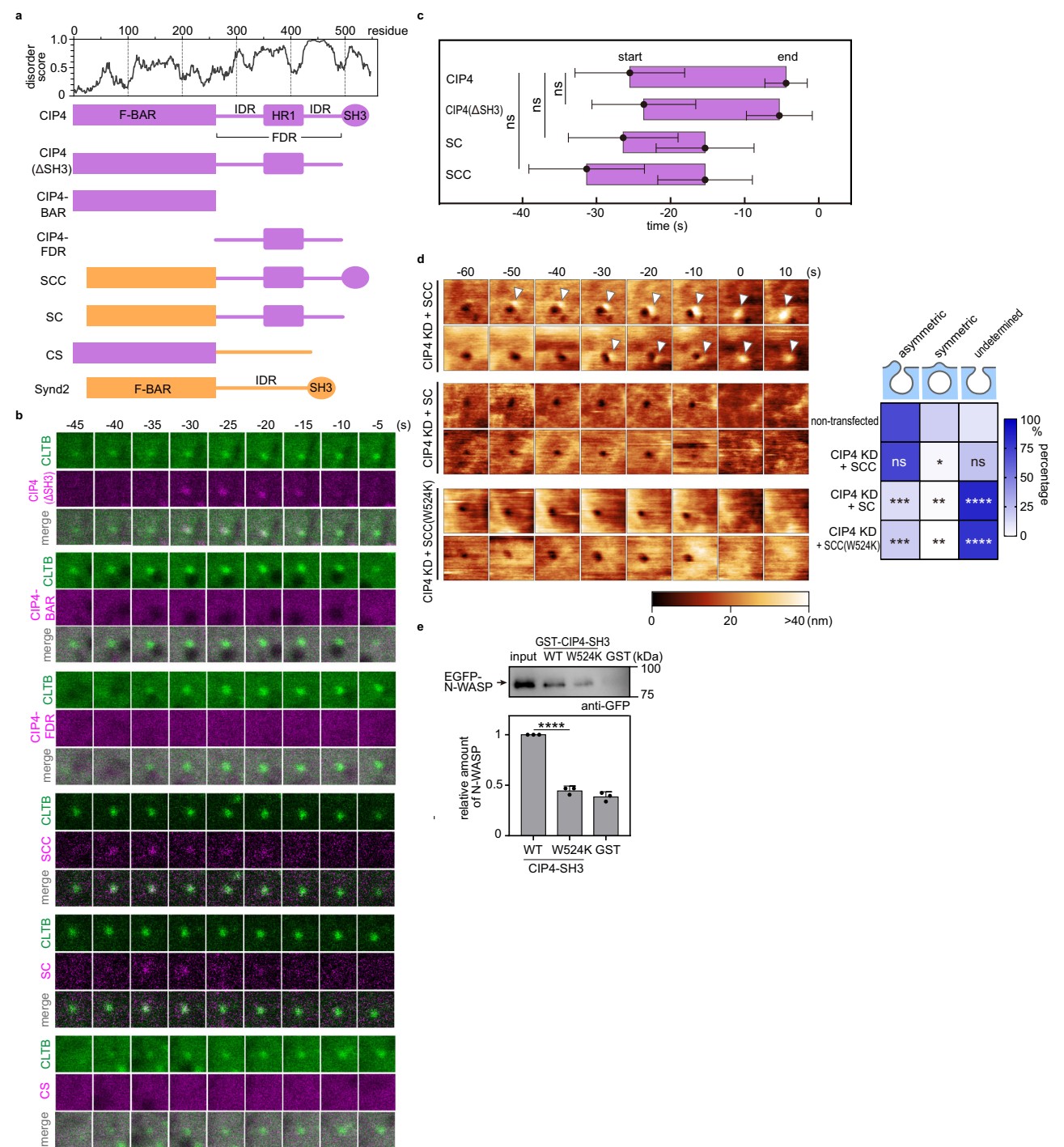

MGD-IST mutation failed to rescue the CIP4-KD phenotype (Fig. 4d). These results indicated that activated Cdc42 recruits CIP4 to the CCP site, which further induces asymmetric bulge formation.

## FDR self-assembly generates 'asymmetric' CIP4 distribution

The interaction between Cdc42 and CIP4 is crucial in CIP4 assembly at the CCP; however, it does not explain how 'asymmetricity' is produced. A possible mechanism could be the cooperativity of the assembly. The structural prediction based on the amino acid sequence suggested that the HR1 of CIP4 forms a helical coil composed of two antiparallel α-helices (Supplementary Fig. 6a). To examine the secondary structure of CIP4-HR1, we measured the circular dichroism spectrum of recombinant HR1, which fitted well with that of the reference curve of α-helix (Supplementary Fig. 6b), demonstrating a

strong propensity of α-helix in HR1. Therefore, we speculated that it forms a multimer and the association proceeds cooperatively. To test this hypothesis, purified HR1 was subjected to gel-filtration chromatography to estimate the molecular size. We observed that HR1 existed mainly as a dimer, and any multimers could not be detected under a physiological condition (Supplementary Fig. 6c, d). A similar result was obtained by treating purified HR1 with bis(sulfosuccinimidyl)suberate (BS[3]) crosslinkers and analysing using SDS-PAGE. Adding crosslinkers induced an extra band corresponding to the HR1 dimer, but not higher-order multimers (Supplementary Fig. 6e). Thus, these results demonstrate that HR1 multimerisation is not the major driving force of asymmetricity.

Another possibility could be the self-assembly and phase transition of FDR via electrostatic interaction between IDRs. The HR1 domain

**Fig. 3 | Flanking disordered region (FDR) and SH3 domain are necessary for asymmetric bulge formation. a** (Top) Disorder plot of CIP4 predicted using IUPred. (Bottom) Schematic illustration of the domain structures of full-length CIP4 and Synd2, together with their chimeric constructs. **b** Time-lapse fluorescence images obtained from Cos7 cells expressing EGFP-fused CLTB and mCherry-fused CIP4 (WT or chimeric constructs). The image size is $1.0 \times 1.0\ \mu m^2$. Time 0 was when the clathrin signal disappeared. Representative images are presented here. Additional cases are illustrated in Supplementary Fig. 4a. **c** A summary of the assembly profile of CIP4, CIP4(ΔSH3), and chimeric molecules at CCP. Time 0 was when the clathrin signal disappeared. The timing when the fluorescence signal appeared and disappeared at the CCP area was defined as 'start' and 'end', respectively, and are plotted (N = 20 pits examined over three independent experiments, for each condition). P values were calculated using the two-tailed Student's t test, ns: not statistically significant, $\alpha = 0.05$. The exact P values are provided in Source Data. The error bars represent the standard deviation. **d** Time-lapse HS-AFM images obtained from Cos7 cells expressing EGFP-fused SCC, SC, or SCC(W524K) mutants under the CIP4-KD background. Representative images are presented here. Images were

taken every 10 s. Time 0 was when the pit completely closed on the HS-AFM image. The arrowheads indicate the membrane bulge, and the height information of the AFM image is presented using a colour bar. The image size is $1.0 \times 1.0\ \mu m^2$. The frequency of three closing patterns was measured and is summarised in the right panel (N = 4 biologically independent cells, for each condition). P values were calculated by comparing the ratio of the same closing pattern between the non-treated and treated group using a two-tailed Student's t test and indicated in each block, *P < 0.05; **P < 0.01; ***P < 0.001; ****P < 0.0001; ns: not statistically significant, $\alpha = 0.05$. The exact P values are provided in Source Data. **e** Pull-down assay of CIP4-SH3 and N-WASP. GST-tagged CIP4-SH3(WT) or CIP4-SH3(W524) was mixed with the HeLa cell lysate containing EGFP-fused N-WASP and GSH bead. The eluted fractions were analysed using western blotting with an anti-GFP antibody. The immunoreactive bands were quantified and summarised (bottom). Data are presented as mean ± standard deviation from three independent experiments relative to WT. All data points are shown. P values were calculated using the two-tailed Student's t test, ****P < 0.0001, $\alpha = 0.05$. The exact P values are provided in Source Data.

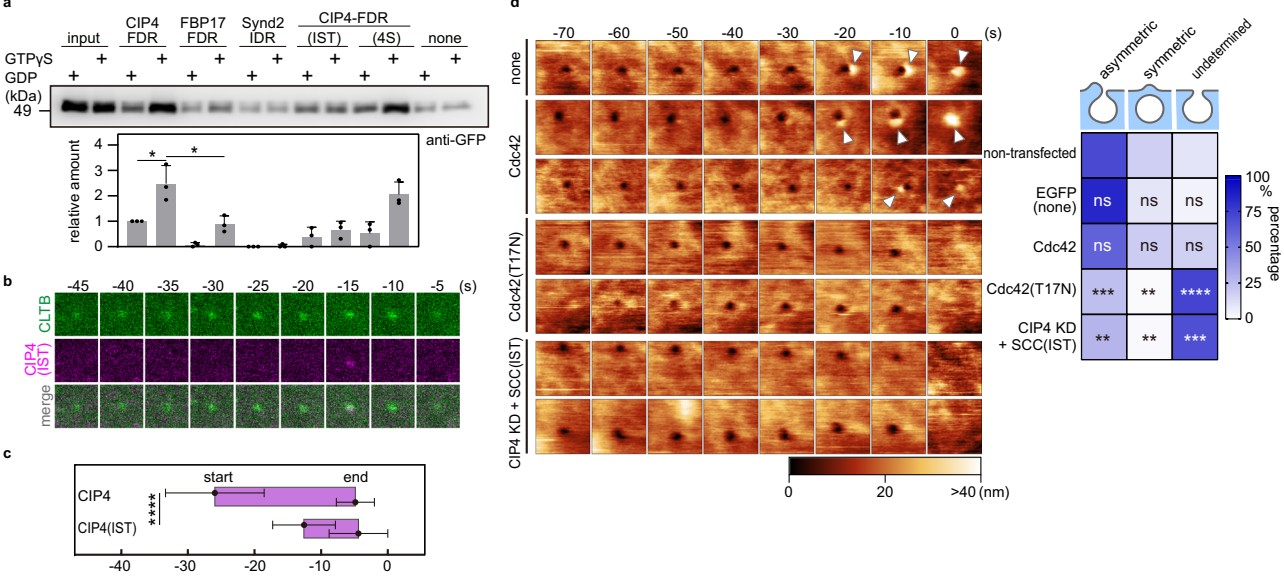

**Fig. 4 | Interaction between Cdc42 and CIP4 is required for the asymmetric bulge. a** Pull-down assay between Cdc42 and CIP4-FDR, FBP17-FDR, Synd2-IDR, CIP4-FDR(IST), or CIP4-FDR(4S) in the presence of GDP or GTP-γS. Hexa-histidine-tagged recombinant proteins were mixed with the cell lysate containing EGFP-fused Cdc42 in the presence of GDP or GTP-γS. The eluted fraction was subjected to the western blot and is summarised in the bottom panel. Data are presented as mean ± standard deviation from three independent experiments as a relative value to the binding amount between CIP4-FDR and GDP-loaded Cdc42. All data points are shown. P values were calculated using the two-tailed Student's t test, *P < 0.05, $\alpha = 0.05$. The exact P values are provided in Source Data. **b** Time-lapse fluorescence images of EGFP-fused CLTB and mCherry-fused CIP4 containing MGD to IST mutation expressed in Cos7 cells. The image size is $1.0 \times 1.0\ \mu m^2$. Time 0 was when the clathrin signal disappeared. Representative images are presented here. Additional cases are presented in Supplementary Fig. 5c. **c** A summary of CIP4 and CIP4(IST) assembly profiles at CCPs. The timing when the fluorescence signal appeared and disappeared at the CCP area was defined as 'start' and 'end',

respectively, and are plotted (N = 20 pits examined over three independent experiments, for each condition). P values were calculated using the two-tailed Student's t test, ****P < 0.0001, $\alpha = 0.05$. The exact P values are provided in Source Data. Error bars represent the standard deviation. **d** Time-lapse HS-AFM images obtained from Cos7 cells expressing EGFP (none), EGFP-fused Cdc42 (WT or T17N mutant), and the cells expressing EGFP-fused SCC(IST) under the CIP4-KD background. Representative images are presented. Images were taken every 10 s. Time 0 was when the pit completely closed on the HS-AFM image. Arrowheads indicate the membrane bulge, and the height information of the AFM image is presented using a colour bar. The image size is $1.0 \times 1.0\ \mu m^2$. The frequency of three closing patterns was measured and is summarised in the right panel (N = 4 biologically independent cells, for each condition). P values were calculated by comparing the ratio of the same closing pattern between the non-treated and treated group using a two-tailed Student's t test and indicated in each block, **P < 0.01; ***P < 0.001; ****P < 0.0001; ns: not statistically significant, $\alpha = 0.05$. The exact P values are provided in Source Data.

of CIP4 and FBP17 is flanked by positively and negatively charged IDRs at the amino and carboxyl sides, respectively (Fig. 5a, Supplementary Fig. 6f). By contrast, the charge segregation of Synd2 IDR is not so prominent (Supplementary Fig. 6f). It has been well-established that a polymer with segregated charges (i.e., block polyampholyte) exhibits stronger liquid–liquid phase separation (LLPS) than the chain with the same number of charges randomly distributed (i.e., random polyampholyte)[37,38], and the pattern of charge blocks is an important

determinant of LLPS properties[39]. These properties strongly suggest that the FDR self-assembles to undergo LLPS via charge-block interaction among IDRs (Fig. 5a). To test this possibility, the purified recombinant protein was subjected to an in vitro droplet assay in the presence of polyethene glycol as a crowder. CIP4-FDR formed protein droplets, unlike the IDR of Synd2 (Fig. 5b). Quantification of protein droplets based on the turbidity revealed that the saturation concentration of CIP4-FDR was similar to that of the FBP17-FDR but

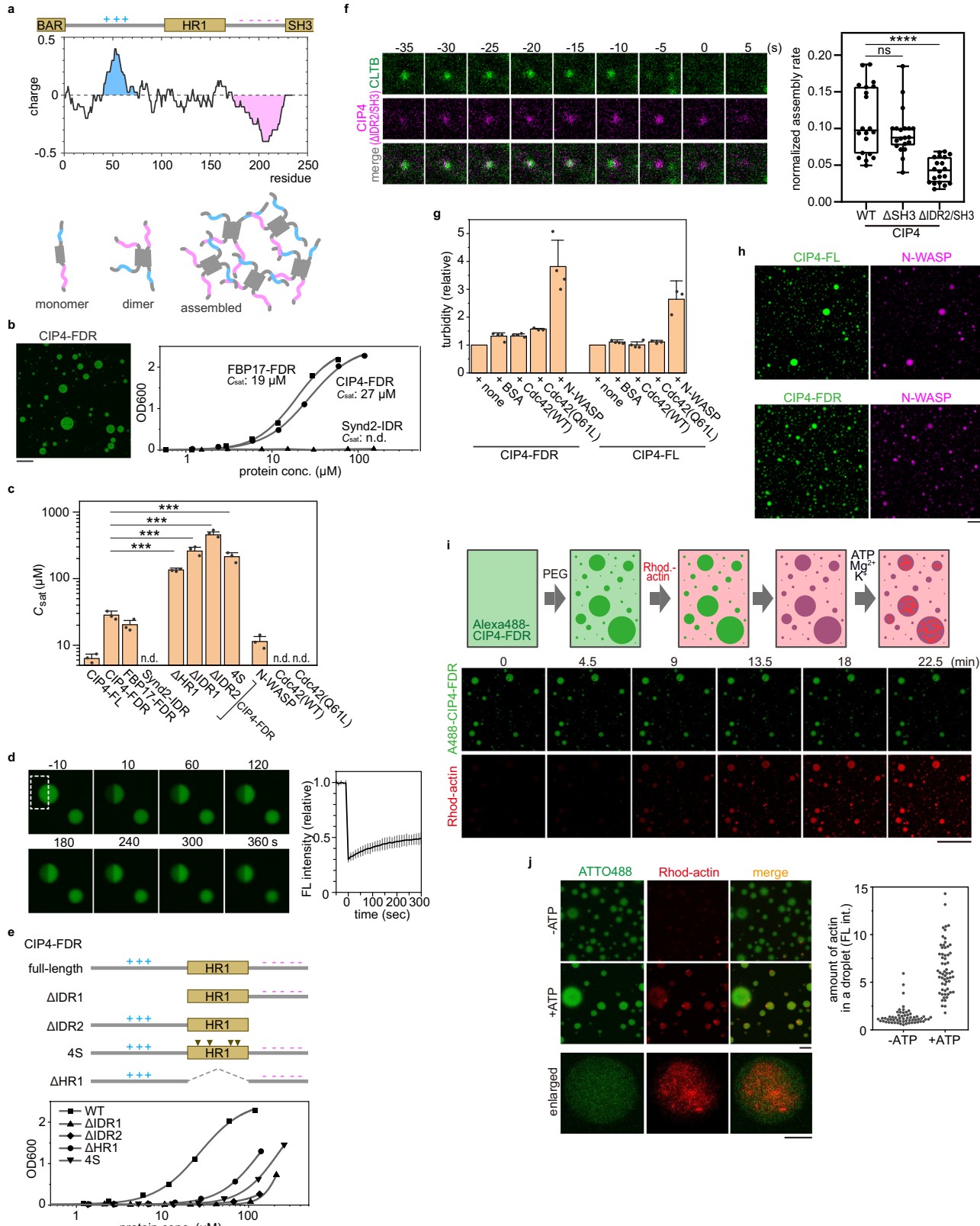

remarkably lower than that of the Synd2-IDR (Fig. 5b, c), indicating that CIP4-FDR and FBP17-FDR has a stronger propensity for LLPS than Synd2-IDR. Fluorescence recovery after photobleaching (FRAP) analysis revealed that the protein droplet was in a liquid-like state (Fig. 5d). Deletion of either IDR (ΔIDR1 or ΔIDR2) from CIP4-FDR severely attenuated the LLPS (Fig. 5c, e), demonstrating that the interaction between charge blocks is essential in the LLPS of FDR.

Deleting one of the IDRs of CIP4-FDR did not completely abolish LLPS (Fig. 5e). Therefore, we speculated that HR1 also contributes to LLPS. HR1 deletion from FDR (ΔHR1) partly, but significantly, reduced the LLPS propensity (Fig. 5c, e). To substantiate this idea, intensive mutations were introduced into HR1 to reduce dimer formation. We observed that substitutions of four hydrophobic residues (L347, L401, Y408, and W411 labelled in Supplementary Fig. 6a) by serine (4S)

**Fig. 5 | CIP4-FDR phase separates and promotes actin polymerisation. a** (Top) A charge plot of CIP4-FDR. HR1 is flanked by net positive (marked in blue) and negative (marked in red) IDRs. (Bottom) A schematic illustration of phase separation of CIP4-FDR driven by charge block interaction. **b** In vitro phase separation of FDRs from CIP4 and FBP17 and IDR from Synd2. The turbidity of the solution was measured based on the optical density at 600 nm and plotted against the protein concentration. The saturation concentration ($C_{sat}$) was obtained via curve fitting (Methods) and is indicated. A representative microscopic image of the CIP4-FDR droplet is also presented. Similar results were obtained from three independent experiments. For fluorescence imaging, the protein was pre-labelled with ATTO488. The scale bar is 50 μm. **c** The summary of in vitro phase separation assay. The $C_{sat}$ was obtained from the turbidity assay and is summarised. *P* values were calculated using the two-tailed Student's *t* test, ***$P < 0.001$, $\alpha = 0.05$. The exact *P* values are provided in Source Data. Error bars represent the standard deviation from three independent experiments. All data points are shown. n.d. not determined. **d** FRAP analysis of a protein droplet formed by CIP4-FDR. The protein droplet of ATTO488-labelled CIP4-FDR was prepared as indicated in (**b**). Half of the target droplet (dotted square) was bleached via laser radiation, and time-lapse imaging was continued every 10 s. Some representative images are presented. The average fluorescence intensity of the bleached region was quantified and plotted against time as a value relative to that of the pre-bleached signal. Data are presented as mean ± standard deviation from three independent measurements. **e** (top) A schematic illustration of CIP4-FDR and its deletion mutants. (bottom) The turbidity assay was performed and plotted as described in (**b**). **f** (Left) Time-lapse fluorescence images of EGFP-fused CLTB and mCherry-fused CIP4(ΔIDR2/SH3) expressed

in Cos7 cells. The image size is $1.0 \times 1.0\ \mu m^2$. Time 0 was when the clathrin signal disappeared. Additional cases are indicated in Supplementary Fig. 6g. (Right) Assembly rate of CIP4 at CCPs was plotted ($N = 20$ pits examined over three independent experiments, for each condition). The median and lower and upper quartiles are indicated in the box. The minimum and maximum values are indicated with upper and lower whiskers, respectively. *P* values were calculated using the two-tailed Student's *t* test, ****$P < 0.0001$; ns: not statistically significant, $\alpha = 0.05$. The exact *P* values are provided in Source Data. **g** The self-assembly of CIP4 is not affected by Cdc42. The relative turbidity of CIP4 (FDR and FL) was measured by the turbidity assays and summarised. Data are presented as mean ± standard deviation from three independent experiments. All data points are shown. **h** Representative microscopic images of the co-separation of CIP4-FL or CIP4-FDR and N-WASP. Similar results were obtained from three independent experiments. For fluorescence imaging, the protein was pre-labelled with ATTO488 or ATTO610. Scale bar, 50 μm. **i** (Top) Schematic illustration of actin polymerisation in a protein droplet of CIP4-FDR. After forming a droplet of ATTO488-labelled CIP4-FDR, rhodamine-labelled G-actin was incubated with ATP, $Mg^{2+}$, and $K^+$. (Bottom) Time-lapse fluorescence observation of CIP4-FDR (ATTO488-labelled) and G-actin (rhodamine-labelled). Scale bar, 100 μm. **j** (Left) A mixture of rhodamine-labelled and non-labelled G-actin (1:9) was loaded with or without ATP and $Mg^{2+}$ and incubated with a pre-formed CIP4-FDR droplet. Representative fluorescence images are presented here. Enlarged images in the presence of ATP are shown below. Scale bar, 10 μm. (Right) Fluorescence intensity of F-actin in the droplet was quantified and plotted ($N = 60$ in each condition).

disrupted the HR1-dimer without affecting the interaction with Cdc42 (Fig. 4a, Supplementary Figs. 5a, 6c). As expected, the LLPS propensity of the CIP4-FDR(4S) mutant was lower than that of the WT (Fig. 5c, e), indicating that HR1 dimerisation promotes the FDR-dependent assembly of CIP4. To test whether FDR-driven LLPS is essential in CIP4 assembly at the CCP, we analysed the assembly rate of CIP4 through live-cell fluorescence imaging. SH3 deletion (CIP4(ΔSH3)) did not affect the assembly rate of CIP4 at CCP (Fig. 5f). Interestingly, further deletion of IDR2(ΔIDR2/SH3), which reduces the LLPS propensity of CIP4-FDR (Fig. 5c, e), reduced the assembly rate by ~70% (Fig. 5f, Supplementary Fig. 6g), implicating the role of phase separation in driving CIP4 assembly at CCP. Similarly, the 4S mutation in the full-length CIP4 almost completely abolished the assembly (Supplementary Fig. 6h), which was more severe than the IDR2 deletion. In agreement with this finding, the SCC chimaera carrying the 4S mutation did not rescue the CIP4-KD (Supplementary Fig. 6i). These observations indicated that the FDR-dependent LLPS is critical in CIP4 assembly at the CCP.

As we demonstrated, CIP4 assembly is closely associated with Cdc42 and N-WASP. We then investigated whether these two regulators or effectors potentially affected the LLPS propensity of CIP4. Cdc42 (WT and constitutively active form Q61L) did not undergo LLPS or enhance CIP4-FDR LLPS (Fig. 5c, g). These results demonstrate that the Cdc42-CIP4 interaction does not contribute to the LLPS of CIP4. However, N-WASP exhibited strong LLPS ($C_{sat}$: ~10 μM) (Fig. 5c). Therefore, we examined whether N-WASP and CIP4 co-exist in the same droplet or exclude each other. N-WASP and full-length CIP4 co-existed in the same droplet (Fig. 5h). The same result was obtained with CIP4-FDR and N-WASP (Fig. 5g, h). These results indicate that both proteins have a strong propensity for LLPS and tend to co-exist in the same droplet.

### Actin amasses and polymerises in a liquid phase of CIP4-FDR

We examined whether a liquid phase formed by CIP4-FDR could accelerate actin polymerisation. Alexa488-labelled CIP4-FDR was mixed with polyethene glycol to induce LLPS, and then rhodamine-labelled G-actin was added (Fig. 5i). The time-lapse observation under a fluorescence microscope revealed that actin rapidly moved into the droplet (Fig. 5i). The concentration of actin was ~100 times higher in the droplet than in the external medium, demonstrating that actin accumulated in the CIP4-FDR droplet.

We then tested whether actin polymerisation also occurs in the droplet. ATP and $Mg^{2+}$ and a low concentration of G-actin (45 μg/mL) were added to the droplet solution of CIP4-FDR to induce actin polymerisation in the droplet (Fig. 5i, j). Under this condition, actin polymerisation rarely occurred in the diluted phase, owing to the low actin concentration. However, fibrous actin was frequently observed in the droplet (Fig. 5j). ATP depletion almost completely abolished it (Fig. 5j), demonstrating that actin polymerisation occurs in the droplet. Thus, these results demonstrate that CIP4 self-assembles via FDR to form a liquid-like phase and concentrates G-actin in the liquid phase to accelerate the polymerisation.

## Discussion

Herein, we used various live-cell imaging techniques and biochemical approaches to demonstrate that the BAR domain-containing protein CIP4 is critical in the actin-dependent asymmetric closing process of CME. CIP4 interacts with curved membrane and actin-related proteins such as Cdc42 and N-WASP through its distinct structured domains (i.e., BAR, HR1, and SH3). Our results demonstrated that the asymmetricity was generated during the self-assembly of the disordered regions between those structured domains. This describes an intriguing molecular mechanism of how a multi-domain protein, such as CIP4, spatiotemporally recruits and orchestrates the actin machinery at a specific cell position. Strong self-assembly, phase separation of IDRs, and stereospecific interactions between structured domains induce a local burst of branched actin fibres nearby the CCP. It has been proposed that unstructured amino acid chains sense the curved lipid bilayer via entropic or electrostatic mechanisms, determined by their length and net charge[40]. Recently, the IDRs of membrane-bound, -anchored, and -embedded proteins, including AP180, Amphiphysin 1, FBP17, CALM, and Epsin, are gaining more research attention for supporting or enhancing the membrane-bending or curvature-sensing properties of the structured domains[41–45]. Therefore, our findings provide a novel perspective for understanding the role of disordered proteins in CME progression.

Four members of F-BAR domain-containing proteins are CIP4, FBP17, Synd2, and Toca1. CIP4, FBP-17, and Synd2 assemble at the CCP at different times during CME (Fig. 2b) and have different cellular functions[32,46] despite their similar domain structures. This could partly be attributed to the slightly different curvature of the three-dimensional structures of the BAR domains[46]. Herein, we demonstrated that the

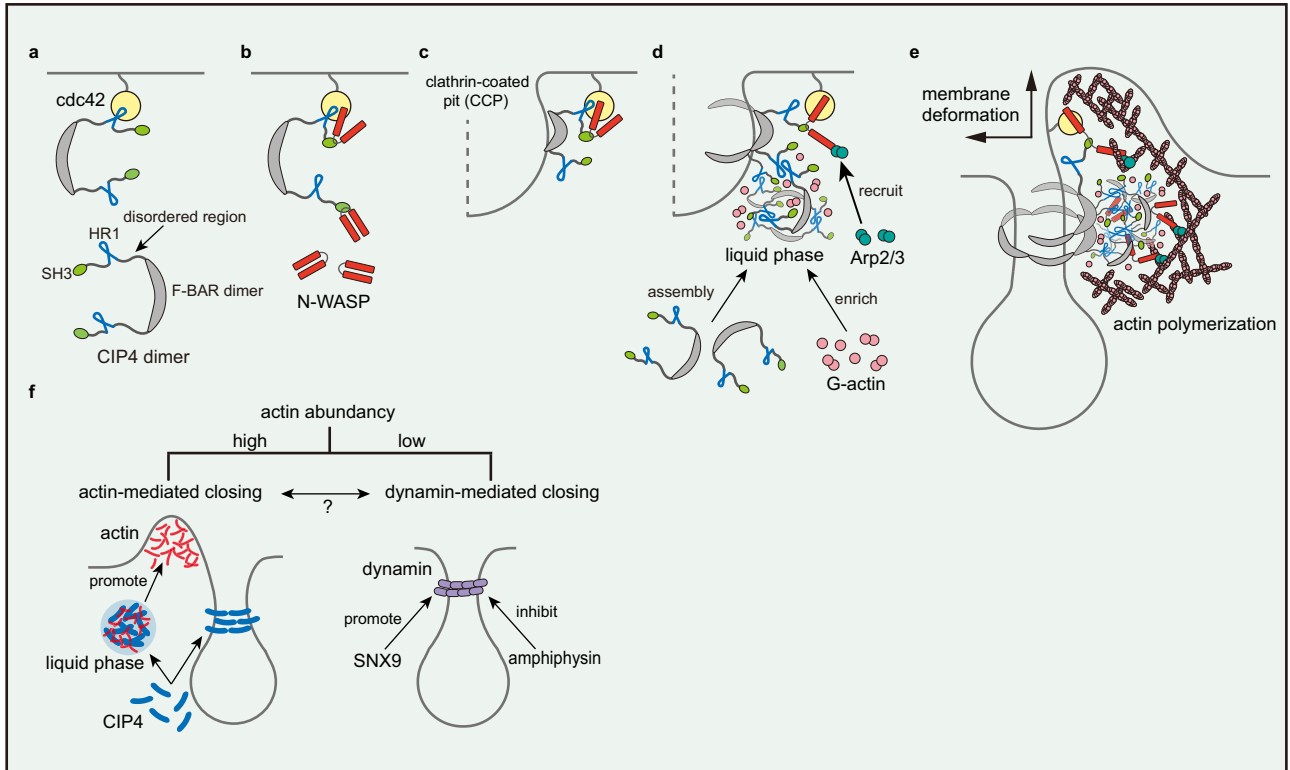

**Fig. 6 | CIP4 drives the asymmetric CCP closing. a–e** Schematic illustrations of the CIP4-induced asymmetric CCP closing. Membrane-bound and activated Cdc42 recruits CIP4 via direct interaction with HR1 (**a**). CIP4 recruits and activates N-WASP to form a ternary complex (**b**). When the membrane-bound complex approaches a CCP, the F-BAR domain of CIP4 recognises and assembles on the curved membrane (**c**). This increases the chance of interaction between FDRs and initiates phase separation, resulting in an asymmetric assembly nearby the CCP (**d**). Meanwhile, activated N-WASP further recruits and activates the Arp2/3 complex, initiating actin polymerisation (**d**). The condensed phase of CIP4-FDR concentrates G-actin and accelerates the local actin polymerisation, which creates an actin-rich environment (**d, e**). Activated N-WASP may co-separate with CIP4 in the condensed phase, accelerating Arp2/3 recruitment. Consequently, polymerised actin accumulated rapidly and asymmetrically around CCP which deforms the plasma membrane and promotes the complete closing of the CCP (**e**). **f** Schematic illustration of the potential correlation between actin- and dynamin-driven pit-closing.

middle-disordered regions between the BAR and SH3 domains had different self-assembly properties. The FDRs of CIP4 and FBP17, but not the IDR of Synd2, had high LLPS propensity, and both disordered regions were required for CIP4 phase separation (Fig. 5c, e). However, when either of the disordered regions was deleted (ΔIDR1 or ΔIDR2), a certain level of LLPS propensity was preserved (Fig. 5e), suggesting that HR1 dimerisation may also promote the LLPS of FDR. These results clearly explain the functional diversity among Synd2, CIP4, and FBP17 in CME and possibly other cellular processes in which they are involved.

In contrast to Synd2, the FBP17-FDR has a similar charge distribution pattern to that of CIP4 (Supplementary Fig. 6f) and exhibits a similar phase behaviour in vitro (Fig. 5b, c); however, they greatly diverge regarding their functions and localisations. During CME, CIP4 assembles at the CCP before complete pit-closure (Fig. 2d), which is remarkably earlier than that of FBP17 (Fig. 2b). Moreover, an in vitro pull-down assay revealed that CIP4-FDR had a higher ability to bind to Cdc42 than does the FDR of FBP17 (Fig. 4a, Supplementary Fig. 5a); this implies that disordered regions have a role in Cdc42 interactions. In dendritic neurons, FBP17 displayed an intracellular distribution different from that of CIP4 due to its poly PxxP region within the first IDR[47]. These observations implicate the underestimated role of disordered regions in regulating nearby stereospecific interactions.

CIP4 strongly self-assembled and underwent LLPS in vitro (Fig. 5b); nonetheless, the intracellular concentration of endogenous CIP4, estimated from the total intracellular amount (~20 nM, Supplementary Fig. 6j), was far lower than the saturation concentration (~6 μM, Fig. 5c). These results suggested that in addition to the charge block-driven LLPS of FDR, a certain mechanism which assembles CIP4 near the CCP for 'nucleation' is required to initiate LLPS. One important factor in this nucleation can be stereospecific interactions involving the structured domains (F-BAR, HR1, and SH3).

In the absence of the CCP, CIP4 binds to active Cdc42 anchored to the plasma membrane via PIP₂ through the HR1 domain (Fig. 6a). CIP4 also binds to N-WASP via the SH3 domain, allowing the interaction between N-WASP and Cdc42, and ternary complexes form among CIP4, N-WASP, and Cdc42 (Fig. 6b). However, at this stage, the local concentration of CIP4 may not be sufficient to initiate LLPS (Fig. 6b). When the ternary complex occasionally approaches the CCP, the BAR domain recognises the curvature of the pit membrane and binds to it (Fig. 6c). The BAR domain multimerises on the membrane[48]; therefore, it accelerates CIP4 assembly near the CCP. In good agreement with this finding, the LLPS propensity of full-length CIP4 is higher than that of FDR (Fig. 5c), suggesting the involvement of the BAR domain in LLPS progression.

Recent studies reported 'membrane-supported' LLPS at the plasma membrane and other intracellular membrane organelles[49,50], where membrane-anchored or -inserted proteins assemble on the membrane and 'nucleate' for LLPS. This two-dimensional system can considerably accelerate LLPS compared to a dispersed system and initiate LLPS regardless of the bulk (cytoplasmic) protein concentration being below the threshold. We speculate that these stereospecific interactions (Cdc42-HR1, F-BAR-F-BAR, and F-BAR-membrane) contribute to CIP4 nucleation near the CCP.

Once CIP4 nucleates near the CCP, it undergoes 'charge block-driven LLPS' via FDR and condenses—a major driving force of 'asymmetric' CIP4 assembly (Fig. 6d). The F-BAR domain multimerises and

forms a helical structure around the tubulated membrane in vitro[48,51]. However, it is insufficient for the assembly at the CCP, as we have demonstrated in this study (Fig. 3b). Furthermore, it does not provide a structural basis for the 'asymmetric' distribution at the CCP. Instead, we speculate that LLPS is key to forming the 'asymmetric' assembly of CIP4 because the emergence of a condensed phase proceeds in a non-continuous manner when the local protein concentration is above the threshold. Therefore, F-BAR- and HR1-driven nucleation and FDR-driven LLPS synergistically work for the fast assembly of CIP4 at the CCP. This finding is consistent with the results of a previous study demonstrating that disordered domains enhance the affinity of BAR domains to the curved membrane[41].

Our chimeric molecule between CIP4 and Synd2 demonstrated that the BAR domain was functionally exchangeable (SCC chimaera, Fig. 3a–c, Supplementary Fig. 4a), implying that the curvature-sensing by the BAR domain is not critical during this step of the CME and IDR is more important. The SH3 domain was unnecessary for CIP4 assembly (Fig. 3a–c, Supplementary Fig. 4a). However, it is necessary to recruit N-WASP and subsequently form the asymmetric bulge.

A CIP4-rich condensate near the CCP is then supposed to recruit other related proteins into the condensate (Fig. 6d). Compared to HR1, GBD of N-WASP has a higher affinity to Cdc42 and will completely displace the HR1 to interact with Cdc42, which further exposes the C-terminal acidic region, allowing the further recruitment and activation of the Arp2/3 complex[52]. We demonstrated that N-WASP also had a high LLPS propensity and co-existed with CIP4 in the same droplet in vitro (Fig. 5c, h), suggesting that the CIP4 condensate recruits and accommodates N-WASP. Consequently, the local concentration of the active Arp2/3 complex also increases, which accelerates actin polymerisation (Fig. 6d, e). A similar process had been demonstrated using Toca1, a subfamily member of CIP4 with conserved HR1 and SH3 domains[36,52], suggesting a conserved role of CIP4 subfamily proteins in mediating Cdc42-WASP-Arp2/3 actin signalling. Moreover, recent studies revealed that actin-associated proteins with a long IDR, such as vasodilator-stimulated phosphoprotein and actin-binding LIM protein 1, formed liquid droplets under physiological conditions and promoted actin polymerisation and bundling[53,54]. Our in vitro assay demonstrated that G-actin preferred to localise in the condensed phase of CIP4-FDR (Fig. 5i), implying that the concentration of free G-actin increases in the CIP4-rich phase near the CCP, which also promotes actin polymerisation.

We speculate that 'branched actin', not the CIP4 condensate, generates the mechanical force to deform the plasma membrane. CK666 inhibited actin polymerisation, which consequently inhibited the membrane protrusion (Fig. 1b, Supplementary Fig. 1f), indicating that the major driving force of the protrusion is Arp2/3-mediated actin polymerisation. Electron microscopic observations revealed that branched actin fibres near the CCP are highly anisotropic[17]. However, Arp2/3 produces branched fibres with a fixed angle (~70°). This finding suggests that actin fibres expand in a radial orientation and push the nearby membrane, resulting in membrane bulge (upward) and fusion (lateral) (Fig. 6e). It may also generate a downward force to push the CCP toward the cell interior, as suggested by another group[55]. Further studies are required to elucidate this mechanism.

The BAR/F-BAR scaffolds interplay with dynamin during the membrane scission process[56]. One classic example is the opposite role of Sorting nexin 9 (SNX9) and Amphiphysin in regulating the activity of dynamin at the late stage of endocytosis. SNX9 recruits dynamin and promotes the assembly-stimulated GTPase activity of dynamin by stabilising the dynamin-membrane interaction[57,58] (Fig. 6f). By contrast, Amphiphysin is recruited by dynamin and inhibits the dynamin ring formation by destabilising the dynamin-membrane interaction[59,60] (Fig. 6f). Current evidence is not yet sufficient to illustrate the functional interplay between CIP4 or CIP4-mediated actin polymerisation and dynamin. Nevertheless, we have

observed that dynamin depletion[27] and actin inhibition did not completely abolish CME progression (Fig. 1b, Supplementary Fig. 1f), suggesting that actin- and dynamin-driven pit-closing may function as relatively independent mechanisms and either could be dominant depending on free actin abundance (Fig. 6f). However, membrane tabulation induced by the F-BAR domain could be antagonised by dynamin- and membrane-associated actin cytoskeleton[61]. Moreover, each dynamin helix can capture 12–16 actin filaments and align them into bundles[62]. Therefore, it is also reasonable to propose that dynamin may further potentiate the CIP4-induced actin cytoskeleton reorganisation, which promotes dynamin-dependent scission processes such as the super twist of the dynamin helix[63]. However, these speculations should be further studied.

## Methods
### Plasmids
Mammalian expression vectors encoding pEGFP-N1-CLTB and pmCherry-N1-CLTB were constructed by amplifying the fragments of full-length CLTB (amino acids (a.a.) 1–229) using PCR and subcloning into pEGFP-N1 and pmCherry-N1 plasmids respectively. Fragments of full-length (FL) CIP4 (a.a. 1–547), FL-FBP17 (a.a. 1–617), FL-Synd2 (a.a. 1–486), CIP4-F-BAR (a.a. 1–257), CIP4-FDR (a.a. 258–489), CIP4(ΔSH3) (a.a. 1–489), and CIP4(ΔIDR2/SH3) (a.a. 1–425) were amplified using PCR and subcloned into pmCherry-C1 vectors. Fragments of CIP4-FDR (a.a. 258–489), FBP17-FDR (a.a. 258–552), Synd2-IDR (a.a. 276–428), CIP4-FDR-ΔIDR1 (a.a. 339–489), CIP4-FDR-ΔIDR2 (a.a. 258–425), CIP4-HR1 (a.a. 339–425), CIP4-SH3 (a.a. 490–547), and N-WASP-GBD (a.a. 202–259) were amplified using PCR and subcloned into pET28a (+) or pGEX-6P-1 vectors for expression in Escherichia coli. CIP4-FDR(ΔHR1) fragment was obtained by amplifying and ligating the DNA fragments corresponding to CIP4-IDR1 (a.a. 258–338) and CIP4-IDR2 (a.a. 426–489) using PCR.

Vectors encoding mCherry-fused and EGFP-fused SCC chimeric molecules were constructed by amplifying and ligating the DNA fragments corresponding to Synd2 (a.a. 1–275) and CIP4 (a.a. 258–547) and subcloning into pEGFP-C1 and pmCherry-C1 vectors. The S–C region from the SCC fragment was amplified by PCR to obtain the SC chimaera fragments. Similarly, the CS chimaera fragment was obtained by amplifying and ligating the nucleotide sequence corresponding to CIP4 (a.a. 1–257) and Synd2 (a.a. 276–428) using PCR.

Single and multiple amino acid mutations were introduced using PrimerSTAR® Max DNA Polymerase (Takara Bio Inc., Shiga, Japan) according to manufacture's instructions. The MGD to IST mutation was generated by replacing the amino acids of M381, G382, and D383 with I, S, and T, respectively; the 4 S mutation was generated by replacing the amino acids of L347, L401, Y408, and W411 with S; the W524K mutation was generated by replacing the amino acids of W524 with K.

Cdc42 (a.a. 1–191) and N-WASP (a.a. 1–509) fragments were amplified using PCR and subcloned into pEGFP-C1 and pET28a (+) vectors for mammalian and bacterial expression, respectively. pcDNA3-EGFP-Cdc42(T17N) and pcDNA3-EGFP-Cdc42(Q61L) were obtained from Klaus Hahn (Department of Pharmacology, University of North Carolina School of Medicine; Addgene; plasmid # 12600 and 12601)[64]. Primers used in this study are listed in Supplementary Table 1.

### Expression and purification of recombinant proteins
The following recombinant proteins were used herein: hexa-histidine-tagged CIP4-FDR, FBP17-FDR, Synd2-IDR, CIP4-FDR(4 S), CIP4-FDR(IST), CIP4-HR1, CIP4-HR1(4 S), CIP4-FDR(ΔHR1), CIP4-FDR(ΔIDR1), CIP4-FDR(ΔIDR2), Cdc42, Cdc42(Q61L), and N-WASP and glutathione S-transferase (GST)-tagged CIP4-SH3, CIP4-SH3(W524K), and N-WASP-GBD. Vectors encoding the recombinant proteins were constructed as described in the previous section and introduced into the BL21-CondonPlus (DE3)-RIL competent cells (Agilent Technologies, Inc., Santa Clara, CA, USA) via heat shock at 37 °C for 30 s. The expression of

the recombinant protein was induced using 0.5 mM isopropylthio-β-D-galactoside at 16 °C for 16 h. Bacterial cells expressing hexa-histidine-tagged or GST-tagged recombinant proteins were harvested and resuspended with phosphate-buffered saline (PBS; 137 mM NaCl, 2.7 mM KCl, 10 mM $Na_2HPO_4$, and 1.8 mM $KH_2PO_4$; pH 7.4) containing lysozyme (0.2 mg/mL), DNaseI (20 µg/mL), $MgCl_2$ (1 mM), and phenylmethanesulfonylfluoride (1 mM). The cells were subjected to freeze–thaw cycles thrice and homogenised via sonication. Insoluble cell debris was removed via centrifugation at 13,000 g, 4 °C for 15 min. The cleared lysate was then mixed with nickel-nitrilotriacetic acid (Ni-NTA)-agarose beads (QIAGEN, Hilden, Germany) or glutathione (GSH) Sepharose 4B beads (Cytiva, Marlborough, MA, USA), and incubated at 4 °C for 1 h with rotation to collect tagged proteins. The Hexa-histidine- and GST-tagged proteins were eluted with Elution buffer A (100 mM NaCl, 50 mM HEPES, 0.2 mM DTT, and 400 mM Imidazole; pH 7.4) and Elution buffer B (100 mM NaCl, 50 mM HEPES, 0.2 mM DTT, and 200 mM GSH; pH 7.4), respectively, and dialysed against a buffer containing 200 mM NaCl and 50 mM HEPES (pH 7.4) at 4 °C for 6 h. Proteins were concentrated using AmiconUltra (M.W. 3000 Da; Sigma-Aldrich, St. Louis, MO, USA) and stored at −80 °C before use.

## Pull-down assay

The recombinant proteins (~5 µg) were immobilised on ~20 µL of Ni-NTA agarose or GSH Sepharose 4B beads. EGFP-fused Cdc42 and EGFP-fused N-WASP were expressed in HEK293T cells. The cell pellet was resuspended with PBS containing Protease Inhibitor Cocktail (0.5%, Nacalai Tesque, Inc., Kyoto, Japan) and Triton-X-100 (0.25%) on ice for 10 min. The cell lysate containing EGFP-Cdc42 was incubated with 40 mM GTPγ-S or GDP and 20 mM EDTA at 30 °C for 15 min to activate Cdc42[65]. The cell lysate was mixed with the purified recombinant protein and Ni-NTA or GSH beads and incubated at 4 °C for 25 min, and the bound fraction was eluted with 400 mM imidazole or 100 mM GSH in Elution Buffer (100 mM NaCl, 50 mM HEPES, 0.2 mM DTT; pH 7.4), respectively. Approximately 20% of the bound and 10% of the unbound fraction were analysed using immunoblotting and SDS-PAGE, respectively.

## Cell culture and transfection

Monkey kidney-derived fibroblast-like cells (Cos7 cells) purchased from the Cell Engineering Division in RIKEN BioResource Centre (RBC0539; Ibaraki, Japan) were cultured in high-glucose Dulbecco's modified Eagle's medium (DMEM) supplemented with 10% foetal bovine serum at 37 °C with 5% $CO_2$. For HS-AFM observation, slide glasses (Matsunami Glass Ind., Ltd., Osaka, Japan) or 0.15 CG pierced slide glass (ToA Optical Technologies, Inc., Tokyo, Japan) were coated with 0.01% poly-L-lysine solution (Sigma-Aldrich) and incubated at 37 °C for 1 h. Cos7 cells were seeded on the glass, incubated overnight, and subjected to HS-AFM observation the next day. CK666 (Abcam, Cambridge, United Kingdom) was added to the culture medium to a final concentration of 100 µM 30 min before the observation. Polyethyleneimine (M.W. 40,000, Polyscience, Niles, IL, USA) was used to introduce the expression vectors into Cos7 cells.

## RNA interference

Cos7 cells were transfected with siRNA using the HiPerfect reagent (QIAGEN) according to the manufacturer's protocol. siRNA against CIP4 (ID: s17814, 5′-GGAGAAUAGUAAGCGUAAATT-3′), FBP17 (ID: s22916, 5′-CAACCUAAAAAGAACUCGATT-3′), or Synd2 (ID: s22216, 5′-AGAUGUUCUUAAGACCAAATT-3′) were purchased from Ambion Inc. (Austin, TX, USA). siRNA targeting the 3′UTR sequence of CIP4 was customised at Thermo Fisher Scientific (Waltham, MA, USA). siRNA against N-WASP (siGENOME SMART pool) was purchased from Dharmacon Inc. (Lafayette, CO, USA). The cells were treated with trypsin 48 h after the transfection and seeded on slide glasses for HS-AFM observation or directly harvested and analysed using SDS-PAGE

followed by immunoblotting. The rescue experiment was performed by transfecting Cos7 cells with siRNA against CIP4 48 h before the observation and transfecting the cells with vectors encoding mCherry-tagged SC chimaera 24 h before the observation.

## HS-AFM imaging and image analyses

A tip-scan type HS-AFM system (BIXAM™; Olympus Corporation, Tokyo, Japan) combined with a confocal laser scanning microscope (FV1200, Olympus Corporation)[27,66,67] was used for AFM imaging. The HS-AFM system uses a tapping mode with a phase feed-back control. An electron beam-deposited ultra-short cantilever (USC-F0.8-K0.1-T12-x66-10; Nonoworld Corporation, Neuchâtel, Switzerland) with a spring constant of 0.1 N m$^{-1}$ was used for scanning. We performed all imaging using a cantilever amplitude of 85–92% of its free amplitude. Images were acquired every 2 s to analyse the detailed morphological characteristics of the undetermined closing pattern. Images were acquired every 10 s owing to the mutual compensation between scanning speed and scanning area to observe clathrin pits generally. When images were acquired every 10 s, the total scanning area was $6000 \times 4500$ nm$^2$ and was displayed in 320 pixels × 240 pixels images. All observations were performed at 28 °C. To perform the correlative imaging of HS-AFM and confocal microscopy, the autofluorescent cantilever with a fluorescence spot that could be observed by confocal microscopy was immobilised on a holder with a tilted angle of ~12° relative to the x−y plane. The angle between the cantilever tip and the sample stage is nearly 90° allowing the precise identification of the cantilever position and the observation of the cantilever scanning area under the fluorescence microscope. To align HS-AFM images and fluorescence images, the coordinate of the fluorescence spot of the cantilever was set as (0, 0) and the frame rate of both HS-AFM and CLSM was set as 0.1 frame/s once the imaging started.

Time-lapse HS-AFM images were acquired using AFM Scanning System Software Version 1.6.0.12 (Olympus Corporation). The diameter, duration, and height of the membrane bulge were analysed using the Gwyddion software (ver 2.55; http://gwyddion.net/) and GIMP software (ver 2.10.18; https://www.gimp.org/). The highest point of the plasma membrane at the CCP area was recorded and compared with the surrounding plasma membrane to record the height of the membrane bulge. The lowest position in the HS-AFM image was set as '0' automatically. The duration of the membrane bulge was counted from the first image with the visible membrane bulge to its complete disappearance. Regarding the pit's size, its maximum diameter during the entire CME process was measured by first doing a cross-section and measuring the distance between one side of the pit and the other. Concerning cases with no detectable membrane bulge (undetermined pattern), the height and duration were set as '0 µm' and '0 s', respectively. By plotting the change of membrane height at the pit area and pit diameter against time, the slope of the ascending part of the height-time plot was used to obtain the growth rate of the membrane bulge. By contrast, the slope of the descending part of the diameter-time plot was used to obtain the closing rate of CCPs.

## Immunoblotting

The total cell lysate from Cos7 cells expressing EGFP-fused CIP4, FBP17, or Synd2 and from Cos7 cells subjected to single KD of CIP4, FBP17, or Synd2 was separated using 12% SDS-PAGE gel, transferred to a polyvinylidene difluoride membrane, blocked in 5% skimmed milk in tris buffered saline (5 mM Tris, 13.8 mM NaCl, 0.27 mM KCl, pH 7.4) containing 0.1% Tween (Nacalai Tesque, Inc.), and incubated with a primary antibody for 1 h. The following antibodies were used: rabbit anti-CIP4 (1:1000; Bethyl Laboratories Inc., Montgomery, TX, USA), rabbit anti-pacsin2 (1:1000; Novus Biologicals LLC, Centennial, CO, USA), rabbit anti-FNBP1 (1:1000; Novus Biologicals LLC), rabbit anti-N-WASP (1:1000; Cell Signalling Technology, Danvers, MA, USA), rabbit anti-GFP (1:1000; Medical & Biological Laboratories Co., Ltd., Nagano,

Japan), and mouse anti-β-actin (1:1000; Sigma-Aldrich). The blot was then incubated with a secondary antibody: goat anti-rabbit IgG (H + L) antibody (1:5000; Thermo Fisher Scientific) or Goat anti-mouse IgG (H + L) antibody (1:5000; Cytiva) at 27 °C for 1 h. The immunoreactive bands were visualised using Chemi-Lumi One Super Kit (Nacalai Tesque) under a LAS-3000 Imager (Fujifilm, Tokyo, Japan).

### Live-cell fluorescence microscopy
A confocal laser-scanning microscope (FV3000, Olympus Corporation) equipped with a stage incubator (TOKAI HIT Corporation, Shizuoka, Japan) was used for live-cell imaging. The lateral spatial resolution is ~200 nm under optimised operation. All observations were performed at 37 °C, under 5.0% $CO_2$. Cos7 cells were seeded on a 30-mm glass-bottom dish with 40% density and transfected with the vectors mentioned earlier 24 h before imaging. Time-lapse fluorescence images were captured every 5 s and analysed with ImageJ (ver 1.52a; https://imagej.nih.gov) and the GIMP software (ver 2.10.18).

### Super-resolution SIM live-cell imaging
Super-resolution SIM was performed using Elyra 7 with Lattice SIM[2] (Zeiss, Oberkochen, Germany). The lateral spatial resolution was ~60 nm under optimised operation conditions. Regarding two-colour live-cell imaging, Cos7 cells with 40% cell density were seeded on a 30-mm glass-bottom dish and transfected with vectors encoding EGFP-fused CLTB and mCherry-fused CIP4 24 h before imaging. The culture medium in the dish was exchanged with the phenol red-free DMEM (Sigma-Aldrich) 30 min before imaging. Images were taken every 0.5 s and processed with the Zeiss Enz 3.6 blue edition software. The images were analysed with ImageJ (ver 1.52a).

### Circular dichroism (CD) spectroscopy
The CD spectra were acquired with a JASCO J-805 spectropolarimeter (JASCO Corporation, Tokyo, Japan) using 1.0 mL quartz cuvettes with a 5-mm path length. The purified HR1 domain was diluted with 1 × PBS to the final concentration of 0.2 mg/mL. The measurements were performed at 25 °C in the 280–198 nm range with a data point interval of 0.2 nm. The obtained spectra were analysed using the Spectra Manager software (ver 1.0, JASCO Corporation), and the reference spectra were provided by Dr. Yang, J.T[68].

### Crosslinking assay
$BS^3$ crosslinkers (Thermo Fisher Scientific) were dissolved in distilled water, added to the protein solution to a final concentration of 2 mM, and incubated at 28 °C for 30 min. Recombinant His-CIP4-HR1 treated with crosslinkers was subjected to SDS-PAGE and silver staining using Silver Stain II Kit (Fujifilm) according to the manufacturer's instruction.

### Gel-filtration chromatography
Gel-filtration chromatography was performed with the AKTA purifier system (GE Healthcare, Chicago, IL, USA) controlled by the Unicorn 5.1 software. The running buffer contains 20 mM NaCl, 25 mM HEPES, and 0.2 mM 2-mercaptoethanol. Superdex 75 10/300 GL column (Cytiva) with a void volume ($V_o$) of 7.2 mL was used. The calibration curve was obtained by plotting the molecular weight of marker proteins against the partition coefficient ($K_{av}$). $K_{av}$ was calculated with the equation: $K_{av} = (V_e-V_o) / (V_t-V_o)$. $V_e$: elution volume; $V_o$: void volume; $V_t$: total volume.

### Phase separation assay
Hexa-histidine-tagged recombinant proteins of CIP4-FDR, FBP17-FDR, and Synd2-IDR were used for the phase separation assay. The purified protein (~10 mg/mL) was incubated with ATTO488-maleimide (Thermo Fisher Scientific) at 28 °C for 1 h. The reaction was quenched using 4 mM DTT. Droplet formation was induced by adding 5% polyethene glycol and imaged using fluorescence microscopy

(FV3000, Olympus Corporation). For examining interaction with actin, purified actin (Cytoskeleton Inc., Denver, CO, USA) and rhodamine-labelled actin (Cytoskeleton Inc.) were reconstituted with the following buffer: 5 mM Tris-HCl (pH 8.0), 0.2 mM $CaCl_2$, 0.2 mM ATP, and 5% (w/v) sucrose. Rhodamine-labelled and non-labelled actin were mixed at 1:20 to a final concentration of 450 μg/mL and pre-incubated with $Mg^{2+}$ (50 μM) and EGTA (0.2 mM) at 28 °C for 4 min. Purified CIP4-IDR was diluted to 3 mg/mL with the following buffer: 5 mM Tris-HCl (pH 8.0), 1.2 mM ATP, 1.1 mM DTT, 50 mM KCl, 2 mM $MgCl_2$, and 5% (w/v) polyethene glycol (Sigma-Aldrich). $Mg^{2+}$-charged actin was then added to 45 μg/mL and observed using a fluorescence microscope (FV3000).

### Turbidity assay
Recombinant proteins of hexa-histidine-tagged CIP4-FDR, FBP17-FDR, Synd2-IDR, CIP4-FDR-4S, CIP4-HR1, CIP4-HR1(4 S), CIP4-FDR(ΔHR1), CIP4-FDR(ΔIDR1) and CIP4-FDR(ΔIDR2) were used in the turbidity assay. Purified proteins were serially diluted with a Dilution Buffer (50 mM HEPES pH 7.4 and 100 mM NaCl), mixed with an equal volume of Droplet Buffer (30% polyethene glycol, 50 mM HEPES, 100 mM NaCl, pH 8.0), and incubated at 25 °C for 10 min. The mixture was then transferred to a micro-cuvette (BRANDTech Scientific Inc., ESSEX, CT, USA), and the optical density was measured at 600 nm using a spectrophotometer (JASCO V-630). The saturation concentration ($C_{sat}$) was obtained by fitting the data with the following equation using OriginPro (v.9.8):

$$OD_{600} = OD_{max} \, C^n/(C_{sat}^n + C^n),$$

where C represents protein concentration.

### Multiple sequence alignment
Multiple sequence alignment was performed using the T-Coffee test (https://tcoffee.crg.eu/). Sequence similarity was determined using EMBOSS Needle (https://www.ebi.ac.uk/Tools/psa/emboss_needle/).

### Three-dimensional protein structure predictions
The three-dimensional protein structure was predicted using Colab-Fold (v.1.5.2: AlphaFold2 using MMseqs2) (https://colab.research.google.com/github/sokrypton/ColabFold/blob/main/AlphaFold2.ipynb)[69,70]. The amino acids sequence of CIP4-HR1 used for the prediction is shown in Supplementary Fig. 3d.

### Statistical analysis
To analyse the closing patterns of CCPs, relatively equal numbers of CCP images were collected from four different Cos7 cells. The total number of CCPs analysed in each analysis is described in the figure legends. All statistical analyses were performed using the two-tailed Student's $t$ test unless otherwise indicated. Differences were considered significant at $P < 0.05$.

### Reporting summary
Further information on research design is available in the Nature Portfolio Reporting Summary linked to this article.

## Data availability
Amino-acid sequences of *Homo sapiens* CLTB, *Mus musculus* CIP4, *Homo sapiens* FBP17, *Mus musculus* Synd2, *Homo sapiens* Cdc42, and *Homo sapiens* N-WASP can be obtained from UniProt database (accession number P09497 for CLTB, Q8CJ53-3 for CIP4, Q96RU3 for FBP17, Q9WVE8 for Synd2, P60953 for Cdc42, and O00401 for N-WASP). All data supporting the conclusions and findings included in this study are available within the article or Supplementary Information. Source data are provided with this paper. Any additional information required to reanalyse the data reported in this study is available from the corresponding author upon request. Source data are provided with this paper.

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

## Acknowledgements

We thank T. Ozaki, K. Deguchi, S. Dodo, and B. Yi at Kyoto University, Graduate School of Biostudies for their technical assistance. We are especially thankful to Dr. F. Ishidate at the iCeMS Analysis Centre, Institute for Integrated Cell-Material Sciences (WPI-iCeMS), Kyoto University Institute for Advanced Study (KUIAS), for his technical assistance in super-resolution structural illumination microscopy (Elyra 7). We also thank Mr. Sakai, Mr. Uekusa, Mr. Yagi, and Mr. Ito for their assistance in constructing and maintaining the HS-AFM system, and Dr. Y. Sasaki at Kyoto University for their useful discussion on protein multimerisation. This work was supported by JSPS KAKENHI Grant Numbers JP18H02436, JP18KK0196, JP19K22422, 19H04830, JP22H05171, and JP23H00369 to S.H.Y. and AMED (Japan Agency for Medical Research and Development) under Grant Number JP18gm5810018 and JP20wm0325009 to S.H.Y.

## Author contributions

Experiments were designed by Y.Y. and S.H.Y. Plasmids construction, recombinant protein purification, pull-down assay, cell culture and transfection, live-cell HS-AFM imaging and correlative imaging, live-cell confocal fluorescence microscopy, super-resolution SIM live-cell imaging, and gel-filtration chromatography were performed by Y.Y. Phase separation assay and CD spectroscopy were performed by Y.Y. and S.H.Y. Data analysis was completed by Y.Y. and S.H.Y. Manuscript writing, figure design, and editing was done by Y.Y. and S.H.Y. S.H.Y supervised and funded the project.

## Competing interests

The authors declare no competing interests.
