## [Peer Review File · Nature Communications]

Self-assembly of CIP4 drives actin-mediated asymmetric pit-closing in clathrin-mediated endocytosisReviewer #1 (Remarks to the Author):

In the presented manuscript Yiming Yu and co-authors report their study of the closing and scission mechanism of clathrin coated pits (CCP) – the final step of CCP formation that is not well understood. Previously it was shown that actin filaments can assist CCP scission. The authors investigated the role of CIP4 (an F-BAR protein that is thought to promote actin filament formation) in CCP scission using various live-cell imaging techniques and biochemical methods.

In the first part of the study authors describe three patterns of CCP closing based on the topography of the surrounding membrane: asymmetric, symmetric and undetermined. CIP4 appeared to be necessary for the formation of actin-dependent asymmetric bulges.

In the second part of the study authors investigated how different domains of CIP4 contribute to the asymmetric CCP closing and described the interplay between the BAR domain and the disordered regions. Combining the two parts the authors propose the CIP4-driven mechanism of the asymmetric CCP closing.

I find the study timely and important for progressing our understanding of the CCP formation. The study is also methodologically interesting as it correlates the in vivo observations of large biological compartments (CCPs) with more focused in vitro biochemical investigations of protein domains and thus helps bridging the knowledge gap at their interface. I think that this work can be suitable for publication upon fixing some items in the manuscript.

One thing I find interesting to see in the scope of this study:

Assuming that the membrane protrusions in HS-AFM images colocalizes well with clathrin fluorescence signal, it would be interesting to see the angular distance between the asymmetric bulges and the CIP4 signals if the protrusion is placed in the center of coordinates. This would give some primary hints on the spatial orientation of the bulge that could be revisited in the mechanism presented in fig.6. Do authors think that this measurement is relevant considering the resolution difference between fluorescence microscopy and AFM?

Regarding the manuscript itself, below I would like to add some comments that hopefully could help strengthening the presentation of the findings and improving the manuscript readability.

Comments about methods:

1. It can be challenging to image soft surfaces like cells with AFM. I find the quality of these AFM images quite impressive. To improve reproducibility authors could provide more details on AFM imaging:

- a. Specify the AFM imaging mode. I assume they used tapping mode. If this is the case, the rough feedback loop set point (or range of set points) should be mentioned (e.g. 90% of free amplitude).
- b. The authors specified HS-AFM scanning speed per frame and the scanning area in nm². It would be useful to add the resolution in pixels as the AFM scanning rate depends on the number of lines per image (due to x-y piezos extension rates).
- c. Why was the AFM data collected at 10s/frame rate if AFM could image at 2s/ frame without obvious loss in image quality (ext. data fig.1d)? In line 82 authors mention that the fastest events lasted 40 seconds which would be captured only in 4 frames at 10s/frame rate. Maybe the authors could discuss the limitations of their imaging setup.

2. The authors used AlphaFold2 to predict the structure of CIP4-HR1. As far as I know there is no standardized procedure for presenting AlphaFold2 results yet. However, the authors definitely need to provide more details on the simulation. One way could be to provide the exact AlphaFold2 build that was used and the input sequence to make sure that readers could run the same simulation reproducibly. Another way would be to provide the AlphaFold2 output in supplementary (3D coordinates, pLDDT, predicted aligned error).

Comments about the text and the figures:

1. Although I find the authors findings very interesting, the manuscript is quite difficult to read. I suggest authors should polish the text further to make it clearer.

2. The authors use many diverse techniques in this study. It would be good to briefly mention some basic knowledge to make the manuscript accessible for broader audience that may not be familiar with all the techniques used. (For example, the CD spectrum in ext. data fig.6.b does exhibit a strong propensity of α -helix as mentioned in lines 287-288. But it would be good to mention the features of the spectrum that the reader should look at to see this propensity as CD spectroscopy is not commonly used by people who study endocytosis – the potential target audience of this paper.

3. It is not clear what is the difference between "duration" and "lifetime" (lines 95-96, fig 1d).

4. Figures are difficult to read and need to be rearranged:

Fig.1: Legends and axes are too small and difficult to read;

Fig.2: the caption order appears to be a-b-f-c-d-e when following columns or a-c-d-b-f-e when following rows. This is misleading and needs to be rearranged;

Fig.3: the caption order appears to be a-d-e-b-c or a-d-b-e-c; the bottom-left caption does not have a corresponding letter;

Fig.4: Some fonts are too small and difficult to read

Extended Data Fig.2 and 4: captions are not ordered

Other figures are clearer but still could be arranged more carefully and orderly.

5. Supplementary video 1: it would be useful to have a real scale in nm instead of "low" and "high". Also, it would be good to provide videos of all three closing patterns, not just the asymmetric one.

Reviewer #2 (Remarks to the Author):

This manuscript by Yu et al describes the role of the FBAR protein CIP4 in driving asymmetric pit closure during CME in conjunction with NWASP, Arp2/3 and actin. It contains extensive amounts of data of varying types, both in cell and in vitro biophysical data with appropriate statistics quoted and data displayed properly. It is generally well written, although some of the referencing seems overly selective and not appropriately extensive. The discussion would benefit from an extra figure/figure panel and a little more explanation:

1 How does CIP4/actin integrate with the well established roles of dynamin and SNX9/Amphiphysin roles in scission – a figure would greatly help along with an explanation

2 The authors need to explain or at least put forward a possible explanation for why the bulge is asymmetric when FBAR proteins form circular/helical symmetrical structures around templates as seen from the electron microscopy reconstructions eg those by Hinshaw and DeCamilli

The majority of the data presented seem of an acceptable standard, although I think a few controls and techniques are missing, which should be carried out and included

Text

Much work from many labs, especially Haucke, Owen, Traub, Stachowiak, McMahon have shown other proteins are necessary for CME initiation, especially the pioneers FCHO, Eps15 in addition to AP2 – these should be listed and referenced as appropriate, certainly papers published in the last 2-3 years

The asymmetric distribution of actin in mammalian cells was first elegantly demonstrated by Taylor and Merrifield in their seminal 2011 PLoS paper – it should be briefly described and referenced

Syndapin2 is more often called Pacsin2 and this should be stated. A study from the Traub lab

showed that it participates in CME causing phenotypes but not inhibiting endocytosis levels – this may well be explained by the data presented in this paper and should be noted.

Line 120 Do authors mean inhibition of pit closure, which is an easy concept to grasp, when they say promoted formation of CCP

Line 177 – for ease and completeness times from clathrin disappearance as well as closure should be listed on line 177 – changing from one metric to another is confusing

Line 185 – has asymmetric distribution of CIP4 actin etc been actually seen by Taraska lab using their elegant purple false coloured CLEM? If so, as I understand, it needs to be referenced.

Line 192 what is HR1 short for?

Do the disordered regions contain any characterised CME protein binding motifs eg for AP2 Clathrin Eps15 FCHO? This should be stated one way or the other.

Line 279 17% in relation to what please?

Line 294, is phase separation via charged unstructured portions common? A little more explanation of how this phenomenon occurs would be very instructive

Line 376 Please include panel or figure explainaing how CIP4/actin etc relates to amphiphysin/SNX9 BARs and dynamin for scission – it is unclear.

Line 385 CALM (Merrifield/Owen) epsin(deCamilli/McMahon) as well as those stated (these need referencing) primarily use their folded domains for sensing and stabilising or driving membrane curvature – the unstructured parts provide some of the mechanical force by crowding but do not sense the membrane's curvature – they do not touch it. The difference in roles should be carefully stated and explained. Even Stachowiak lab states that both are important and work in tandem.

Technical issues that need to be addressed

What spatial and temporal resolutions are possible using the techniques described – these should be stated

Lines 273-279 This si at best confusing to this reviewer. What was delay of CIP4 assembly inhibited in comparison to wt measured from? Previously two time 0 shave been used – disappearance of clathrin and complete closure. Is this a third measurement? If so a standard one should be used. A value of -12.3 for a mutant si identical to the value for wt CIP4 if it is cloure (line 177). However, Fig4c suggests wt suggests -26s a value identical to that for clathrin disappearance. This need carefully explaining and/or redoing/correcting depending on what the authros actually mean

Line 280 did wt CIP4 rescue the phenotype in comparison? Please state in text. If not know it should be, please do the experiment.

Line 290 and again 348 – if you are going to consider multimerization state gel filtration is not adequate. SEC MALS needs to be sued to avoid Rg artefacts from extended BAR domain dimers etc

Line 354 Why was SH3 not left intact and IDR1, IDR2 and IDR1+IDR2 mutated to properly dissect out functions of IDRs vs SH3? I think it should beif a decent explanation cannot be given.

Line 406 and earlier if you are going to state differences in affinities quantitating pull downs is not acceptable – BLI or SPR is needed to place numbers to be quantitative

Reviewer #3 (Remarks to the Author):

(See attached Word document)

Reviewer #3 Attachment on the following page

Reviewer for Manuscript NCOMMS-22-53719: Self-assembly of CIP4 drives actin-mediated asymmetric pit-closing in clathrin-mediated endocytosis

The authors utilize a combination of high-speed AFM and quantitative fluorescence to elucidate a potential mechanism by which actin polymerization can drive the closing of clathrin coated pits. Their results indicate that CIP4 may be central in regulating this process. In short, key interactions between CIP4, N-WASP, Cdc42 on the plasma membrane leads to recruitment of the Arp2/3 complex which inevitably promotes actin polymerization. This polymerization proceeds into the membrane, bending it outward asymmetrically until fission is completed. The authors arrive at this conclusion by performing live cell experiments in which they knockdown and overexpress various F-BAR containing proteins (along with their mutants) while measuring the topography of the membrane surrounding clathrin-coated pits. In addition, they perform correlative fluorescence imaging that allows them to quantify the lifetimes and colocalization of proteins on or near the pits. Further, they were able to demonstrate that CIP4 has a propensity to form liquid condensates *in vivo*, that actin preferentially partitioned to these condensates, and that actin polymerized within these condensates. When taken together, these results compile a rather fascinating story which details the potential molecular mechanisms associated with a highly relevant physiological process. While this mechanism is speculative (and the author's do an excellent job of acknowledging that), I believe that this work is highly innovative, utilizing advanced techniques to provide new insights to questions that have remained unanswered in the field of membrane remodeling. Therefore, I believe that this manuscript would be of broad interest to readers of Nature Communications since it provides a novel framework for investigating protein-lipid interactions. I recommend this manuscript for publication after the following comments and questions are addressed.

Comments and Questions:

1. As the authors are likely aware, the "constant curvature" and "constant area" models for clathrin-coated pit maturation are highly debated within the field of membrane remodeling. The authors should provide some discussion on how their proposed mechanism fits into or is consistent with either of these models.
2. The authors created an idealized, *in vitro* scenario for liquid-droplet formation with the isolated FDR region of CIP4. However, the actual cytosolic environment is much more complex with a variety of proteins, all with different expression levels within the cell. How does the presence of multivalent binding partners (e.g. N-WASP, Cdc42, etc.) influence the ability of this protein to form liquid droplets? Unless I missed it, the authors do not provide any data to answer this or discuss

this scenario. The authors must either provide data or discuss this scenario in the manuscript.

3. Furthermore, the authors must address whether there is any evidence that the concentrations and stoichiometry of cytosolic proteins is sufficient to induce the type of phase separation that is necessary for their proposed mechanism to be valid.
4. Are the authors implying that droplets were formed with the Δ I_{DR1} and Δ I_{DR2} mutations in Figure 5d? If so, how is that possible with the absence of alternating charge in either mutant? The authors must clarify this in the manuscript.
5. In Figure 4D, the “none” column seems to result in an increased percentage of asymmetric structures when compared to the “non-treated” column of Figure 1B. If I’m understanding correctly, the only difference between these two columns is that Fig. 1B contains untransfected Cos7 cells while Fig. 4D contains Cos7 cells that are overexpressing EGFP. Why is there an increase in asymmetric pits when EGFP is overexpressed?
6. In Fig. 4D, was Cdc42 knocked down in cells? If so, explicitly say so in the figure. If not, why does the expression of EGFP-Cdc42 result in a loss of asymmetric structures and increase in symmetric structures when compared to the scenario where EGFP alone is expressed (“none” column)? Is the EGFP-Cdc42 somehow interfering with the activity of the endogenous Cdc42? The authors must address these concerns in the manuscript.
7. In Figure 6d-e, the authors’ proposed mechanism depicts CIP4 liquid droplets that protrude from the membrane surface. However, the F-BAR domain’s affinity for the membrane would make this protrusion difficult and either confine any droplet formation to the 2-dimensional membrane surface, or bend the membrane inward as described by the mechanism from reference #42 in the manuscript. Assuming that an inward invagination is not formed, the authors must explain either a) how an outward protrusion of liquid droplet is feasible or b) how a planar 2-dimensional droplet can effectively sequester and promote actin assembly.
8. It is not clear how the actin polymerization is polarized in the proposed model. What aspects of this network would actively orient the polymerization in a direction that allows it to bend the membrane and not just extend into the cytosol? The authors must address this question in the manuscript.
9. There is extensive use of the phrase “*in vivo*” in the manuscript, but live cell experiments are not *in vivo*. The authors need to fix the wording to reflect this.

Reviewer #1 (Remarks to the Author):

In the presented manuscript Yiming Yu and co-authors report their study of the closing and scission mechanism of clathrin coated pits (CCP) at the final step of CCP formation that is not well understood. Previously it was shown that actin filaments can assist CCP scission. The authors investigated the role of CIP4 (an F-BAR protein that is thought to promote actin filament formation) in CCP scission using various live-cell imaging techniques and biochemical methods.

In the first part of the study authors describe three patterns of CCP closing based on the topography of the surrounding membrane: asymmetric, symmetric and undetermined. CIP4 appeared to be necessary for the formation of actin-dependent asymmetric bulges.

In the second part of the study authors investigated how different domains of CIP4 contribute to the asymmetric CCP closing and described the interplay between the BAR domain and the disordered regions. Combining the two parts the authors propose the CIP4-driven mechanism of the asymmetric CCP closing.

I find the study timely and important for progressing our understanding of the CCP formation. The study is also methodologically interesting as it correlates the in vivo observations of large biological compartments (CCPs) with more focused in vitro biochemical investigations of protein domains and thus helps bridging the knowledge gap at their interface. I think that this work can be suitable for publication upon fixing some items in the manuscript.

One thing I find interesting to see in the scope of this study:

Assuming that the membrane protrusions in HS-AFM images colocalizes well with clathrin fluorescence signal, it would be interesting to see the angular distance between the asymmetric bulges and the CIP4 signals if the protrusion is placed in the center of coordinates. This would give some primary hints on the spatial orientation of the bulge that could be revisited in the mechanism presented in fig.6. Do authors think that this measurement is relevant considering the resolution difference between fluorescence microscopy and AFM?

Response: We appreciate this comment. We performed the following analyses using the same data set (Fig. 2d).

- i) As the reviewer suggested, the centroid of the protrusion was set as the centre coordinate, and the angular distance between the centroid of the protrusion and the CIP4 signal was determined. The angular distance mostly lies between 90° and 180° without a clear peak (Fig. a).
- ii) Instead of considering the centroid of the protrusion as the centre of the coordinate, the centre of the coordinate was defined as the centroid of CCP in the HS-AFM image and the centroid of the clathrin signal in the fluorescence image (Fig. b). Then, two images were overlaid with the centre of the coordinate, and the distances and the angles of the membrane bulge in the HS-AFM image and CIP4 spot in the fluorescence image were measured. Finally, the angular distance between the bulge and the CIP4 signal was plotted. This analysis provided the angular distance between CIP4 and the membrane bulge relative to the CCP centre. Supposing that CIP4 induces the membrane bulge, the angular distance should be close to 0. However, it did not converge to 0; thus, we could not detect any significant correlation between the CIP4 positions and the membrane bulge (Fig. b).

The most probable reason for this is the lateral resolution of the confocal microscopy (~200 nm under optimal conditions). The distance between the centroids of the pit and the membrane bulge in the HS-AFM image was relatively constant and mostly distributed between 0 and 80 nm (Fig. c). However, the distance between the CIP4 centroids and clathrin spots in the confocal images is broadly distributed between 0 and 180 nm without a clear peak (Fig. d). This contrasts our super-resolution microscope images in Fig. 2e. Therefore, unfortunately, owing to the low x-y resolution of our fluorescence imaging, our correlative images are not durable for analysing the angular distance between CIP4 and the membrane bulge. More reliable conclusions could be drawn if the chance to combine HS-AFM and super-resolution techniques arise in future studies.

Regarding the manuscript itself, below I would like to add some comments that hopefully could help strengthening the presentation of the findings and improving the manuscript readability.

Comments about methods:

1. It can be challenging to image soft surfaces like cells with AFM. I find the quality of these AFM images quite impressive. To improve reproducibility authors could provide more details on AFM imaging:

a. Specify the AFM imaging mode. I assume they used tapping mode. If this is the case, the rough feedback loop set point (or range of set points) should be mentioned (e.g. 90% of free amplitude).

Response: Thank you for your suggestion. Our HS-AFM system uses a tapping mode with a phase feedback control instead of a commonly used amplitude feedback control. This means that it does not have an amplitude set point value. However, we normally scan the cell surface with 85–92% free amplitude. We have revised the Methods section and added this information as follows (page 17, line 443):

‘We performed all imaging using a cantilever amplitude of 85–92% of its free

amplitude.’

b. The authors specified HS-AFM scanning speed per frame and the scanning area in nm². It would be useful to add the resolution in pixels as the AFM scanning rate depends on the number of lines per image (due to x-y piezos extension rates).

Response: Thank you for your suggestion. We agree that this information is necessary. The scanning area of our system is 6 $\mu\text{m} \times 4.5 \mu\text{m}$, and it can be displayed as images of 320 pixels \times 240 pixels. The resolution is $\sim 18.75 \text{ nm}$ per pixel. We have added this information to the method section (page 17, lines 446–447):

‘When images were acquired every 10 s, the total scanning area was $6000 \times 4500 \text{ nm}^2$ and was displayed in 320 pixels \times 240 pixels images.’

c. Why was the AFM data collected at 10s/frame rate if AFM could image at 2s/ frame without obvious loss in image quality (ext. data fig.1d)? In line 82 authors mention that the fastest events lasted 40 seconds which would be captured only in 4 frames at 10s/frame rate. Maybe the authors could discuss the limitations of their imaging setup.

Response: Thank you for your suggestion. The scanning speed and area are in a trade-off; we can achieve a speed of 2 frames/s by sacrificing the scanning area (down to $1 \mu\text{m} \times 1 \mu\text{m}$). Considering the CME frequency in COS7 cells ($\sim 0.022 \text{ pit}/\text{min}/\mu\text{m}^2$, revised Extended Data Fig. 2f), acquiring enough cases using $1 \mu\text{m} \times 1 \mu\text{m}$ scanning area is far more challenging. We have added the explanation in the revised Methods section to clarify this (page 17, lines 444–446):

‘Images were acquired every 2 s to analyse the detailed morphological characteristics of the undetermined closing pattern. Images were acquired every 10 s owing to the mutual compensation between scanning speed and scanning area to observe clathrin pits generally.’

2. The authors used AlphaFold2 to predict the structure of CIP4-HR1. As far as I know there is no standardized procedure for presenting AlphaFold2 results yet. However, the authors definitely need to provide more details on the simulation. One way could be to provide the exact AlphaFold2 build that was used and the input sequence to make sure that readers could run the same simulation reproducibly. Another way would be to provide the AlphaFold2 output in supplementary (3D coordinates, pLDDT, predicted aligned error).

Response: Thank you for your comment. We agree that the detailed procedures for predicting the three-dimensional structure of the CIP4-HR1 should be described. We used ColabFold (v.1.5.2: AlphaFold2 using MMseqs2) (<https://colab.research.google.com/github/sokrypton/ColabFold/blob/main/AlphaFold2.ipynb>) to perform the simulation. The input sequence is identical to that indicated in Extended Data Fig. 3d.

Therefore, we have revised the Methods section (page 21, lines 547–549):

‘The three-dimensional protein structure was predicted using ColabFold (v.1.5.2: AlphaFold2 using MMseqs2 (<https://colab.research.google.com/github/sokrypton/ColabFold/blob/main/AlphaFold2.ipynb>)^{69,70}. The amino acids sequence of CIP4-HR1 used for the prediction is shown in Extended Data Figs. 3d.’

Comments about the text and the figures:

1. *Although I find the authors findings very interesting, the manuscript is quite difficult to read. I suggest authors should polish the text further to make it clearer.*

Response: Thank you for pointing this out. We apologise for our insufficient and unclear descriptions. We have tried to clarify and unify some definitions and also asked professional English editors for suggestions to improve the overall syntax and grammar in the text. We believe that the readability of the revised manuscript has been improved.

2. *The authors use many diverse techniques in this study. It would be good to briefly mention some basic knowledge to make the manuscript accessible for broader audience that may not be familiar with all the techniques used. (For example, the CD spectrum in ext. data fig.6.b does exhibit a strong propensity of α -helix as mentioned in lines 287-288. But it would be good to mention the features of the spectrum that the reader should look at to see this propensity as CD spectroscopy is not commonly used by people who study endocytosis \neq the potential target audience of this paper.*

Response: We appreciate this suggestion. In the revised manuscript, we have added some background information about this technique to the main text and Methods section. The software analysis using curve fitting revealed that the HR1 spectrum almost fits 100 % to the reference curve of α -helix. The revised text read as follows:

pages 7 and 8, lines 178–180

‘To examine the secondary structure of CIP4-HR1, we measured the circular dichroism spectrum of recombinant HR1, which fitted well with that of the reference curve of α -helix (Extended Data Fig. 6b), demonstrating a strong propensity of α -helix in HR1.’

page 19, lines 498–500

‘The measurements were performed at 25 °C in the 280–198 nm range with a data point interval of 0.2 nm. The obtained spectra were analysed using the Spectra Manager software (ver 1.0, JASCO Corporation), and the reference spectra were

provided by Dr. Yang, J.T⁶⁸.’

In addition to the CD spectrum measurement, we have added detailed background knowledge of our phase separation assay and explained why the disordered CIP4 regions could support the liquid phase formation as follows (page 8, lines 191–194):

‘It has been well-established that a polymer with segregated charges (i.e., block polyampholyte) exhibits stronger liquid–liquid phase separation (LLPS) than the chain with the same number of charges randomly distributed (i.e., random polyampholyte)^{37,38}, and the pattern of charge blocks is an important determinant of LLPS properties³⁹.’

We believe the manuscript is now more accessible to various readers.

3. *It is not clear what is the difference between $\dot{E}C$;duration $\dot{E}D$; and $\dot{E}C$;lifetime $\dot{E}D$; (lines 95-96, fig 1d).*

Response: Thank you for this comment. We apologise for not clearly defining these words. Here, ‘duration’ means the duration of membrane bulge, while ‘lifetime’ implies the total lifetime of CCPs. We have added the definition of these words and revised the y-axis in the revised Fig. 1d and the related main text accordingly (page 4, lines 83–86):

‘The asymmetric bulge grew faster (increase in the membrane height per unit time), higher (the maximum height of the bulge), and continued for a longer period (duration of the bulge) than did the symmetric bulge (Fig. 1d). However, CCPs with the asymmetric or symmetric closing patterns had a similar total lifetime (Fig. 1d).’

4. *Figures are difficult to read and need to be rearranged:*

Fig.1: Legends and axes are too small and difficult to read;

Fig.2: the caption order appears to be a-b-f-c-d-e when following columns or a-c-d-b-f-e when following rows. This is misleading and needs to be rearranged;

Fig.3: the caption order appears to be a-d-e-b-c or a-d-b-e-c; the bottom-left caption does not have a corresponding letter;

Fig.4: Some fonts are too small and difficult to read

Extended Data Fig.2 and 4: captions are not ordered

Other figures are clearer but still could be arranged more carefully and orderly.

Response: Thank you for these detailed suggestions. We have made the following revisions:

1. Axis and legends in all the figures have been enlarged if necessary to improve clarity.

2. Captions of all the figures, including those not mentioned in your comment, were reordered following columns or rows. The missing captions were added.

3. We have enlarged and unified the fonts in all the figures. In the revised figures, we have used font size 7 for all the axis, labels, and legends except for *P* values which now have a font size of 5.

5. *Supplementary video 1: it would be useful to have a real scale in nm instead of $\acute{E}C$;low $\acute{E}D$; and $\acute{E}C$;high $\acute{E}D$;.* Also, it would be good to provide videos of all three closing patterns, not just the asymmetric one.

Response: We appreciate these suggestions. In the revised manuscript, we have provided videos for all three closing patterns (Supplementary Video 1–3) with real scale in nm and hope this will improve the manuscript's general integrity.

Reviewer #2 (Remarks to the Author):

*This manuscript by Yu et al describes the role of the FBAR protein CIP4 in driving asymmetric pit closure during CME in conjunction with NWASP, Arp2/3 and actin. It contains extensive amounts of data of varying types, both in cell and in vitro biophysical data with appropriate statistics quoted and data displayed properly. It is generally well written, although some of the referencing seems overly selective and not appropriately extensive. The **discussion** would benefit from an extra figure/figure panel and a little more explanation:*

1 How does CIP4/actin integrate with the well established roles of dynamin and SNX9/Amphiphysin roles in scission \neq a figure would greatly help along with an explanation.

Response: Thank you for your question and suggestion. We agree that discussing the functional relation between CIP4-dependent actin polymerisation is important and informative. Thus, we have proposed some possible scenarios. However, we await direct evidence between actin- and dynamin-dependent closing pathways. Recent studies suggested that actin- and dynamin-driven pit-closing may be relatively independent. Either could dominate depending on the free actin abundance and actin polymerisation. Nonetheless, speculating that CIP4-induced actin polymerisation cooperates with dynamin in membrane scission is reasonable. Therefore, in the revised manuscript, we have added a new section in the Discussion section and a new panel in Fig. 6 (Fig. 6f) to briefly describe these possibilities based on the current understanding of the closing step of CME. Moreover, we also briefly summarised the established role of SNX9 and Amphiphysin in dynamin-dependent scission in this section. Please refer to the main text for the detailed description (pages 13 and 14, lines 338–352):

‘The BAR/F-BAR scaffolds interplay with dynamin during the membrane scission process⁵⁶. One classic example is the opposite role of Sorting nexin 9 (SNX9) and Amphiphysin in regulating the activity of dynamin at the late stage of endocytosis. SNX9 recruits dynamin and promotes the assembly-stimulated GTPase activity of dynamin by stabilising the dynamin-membrane interaction^{57,58} (Fig. 6f). By contrast, Amphiphysin is recruited by dynamin and inhibits the dynamin ring formation by

destabilising the dynamin-membrane interaction^{59,60} (Fig. 6f). Current evidence is not yet sufficient to illustrate the functional interplay between CIP4 or CIP4-mediated actin polymerisation and dynamin. Nevertheless, we have observed that dynamin depletion²⁷ and actin inhibition did not completely abolish CME progression (Fig. 1b, Extended Data Fig. 1f), suggesting that actin- and dynamin-driven pit-closing may function as relatively independent mechanisms and either could be dominant depending on free actin abundance (Fig. 6f). However, membrane tabulation induced by the F-BAR domain could be antagonised by dynamin- and membrane-associated actin cytoskeleton⁶¹. Moreover, each dynamin helix can capture 12–16 actin filaments and align them into bundles⁶². Therefore, it is also reasonable to propose that dynamin may further potentiate the CIP4-induced actin cytoskeleton reorganisation, which promotes dynamin-dependent scission processes such as the super twist of the dynamin helix⁶³. However, these speculations should be further studied.’

2 The authors need to explain or at least put forward a possible explanation for why the bulge is asymmetric when F-BAR proteins form circular/helical symmetrical structures around templates as seen from the electron microscopy reconstructions eg those by Hinshaw and DeCamilli

Response: Thank you for pointing this out. We apologise for our insufficient explanation of the mechanism of the ‘asymmetric’ assembly of CIP4. The F-BAR domain is not the sole major driving force of CIP4 assembly at the CCP; rather, HR1-Cdc42 and BAR-membrane interaction is necessary (Fig. 3b). CIP4 has a strong tendency to self-assemble; therefore, it condenses when the local concentration increases above a critical concentration (saturation concentration). This process is supposed to be a phase transition, and the condensation proceeds in a non-continuous manner. A short paragraph was added to the Discussion section to clarify this (page 12, lines 300–309):

‘Once CIP4 nucleates near the CCP, it undergoes ‘charge block-driven LLPS’ via FDR and condenses—a major driving force of ‘asymmetric’ CIP4 assembly (Fig. 6d). The F-BAR domain multimerises and forms a helical structure around the tubulated membrane *in vitro*^{48,51}. However, it is insufficient for the assembly at the CCP, as we have demonstrated in this study (Fig. 3b). Furthermore, it does not provide a structural basis for the ‘asymmetric’ distribution at the CCP. Instead, we speculate that LLPS is key to forming the ‘asymmetric’ assembly of CIP4 because the emergence of a condensed phase proceeds in a non-continuous manner when the local protein concentration is above the threshold. Therefore, F-BAR- and HR1-driven nucleation and FDR-driven LLPS synergistically work for the fast assembly of CIP4 at the CCP. This finding is consistent with the results of a previous study demonstrating that disordered domains enhance the affinity of BAR domains to the curved membrane⁴¹.’

The majority of the data presented seem of an acceptable standard, although I think a few controls and techniques are missing, which should be carried out and included

Text

1. Much work from many labs, especially Haucke, Owen, Traub, Stachowiak, McMahon have shown other proteins are necessary for CME initiation, especially the pioneers FCHO, Eps15 in addition to AP2 & these should be listed and referenced as appropriate, certainly papers published in the last 2-3 years

Response: Thank you for this comment. We have carefully re-considered individual references and added the following papers from the laboratory studies the reviewer suggested:

5. Zaccai, N. R. *et al.* FCHO controls AP2's initiating role in endocytosis through a PtdIns(4,5)P₂-dependent switch. *Sci. Adv.* **8**, eabn2018 (2022).
6. Day, K. J. *et al.* Liquid-like protein interactions catalyse assembly of endocytic vesicles. *Nat. Cell Biol.* **23**, 366–376 (2021).
7. Lehmann, M. *et al.* Nanoscale coupling of endocytic pit growth and stability. *Sci. Adv.* **5**, eaax5775 (2019).
8. Kadlecova, Z. *et al.* Regulation of clathrin-mediated endocytosis by hierarchical allosteric activation of AP2. *J. Cell Biol.* **216**, 167–179 (2017).
9. Henne, W. M. *et al.* FCHO proteins are nucleators of clathrin-mediated endocytosis. *Science* **328**, 1281–1284 (2010).
10. Yamabhai, M. *et al.* Intersectin, a novel adaptor protein with two Eps15 homology and five Src homology 3 domains. *J. Biol. Chem.* **273**, 31401–31407 (1998).

We have cited these papers in the introduction (page 2, lines 30–32):

‘Clathrin-coated pit (CCP) initiation requires the assembly of several adaptors and scaffold proteins, including the Fes/Cip4 homology Bin/amphiphysin/Rvs (F-BAR) domain-only protein, Eps15, AP2 complex, and intersectin⁵⁻¹⁰.’

2. The asymmetric distribution of actin in mammalian cells was first elegantly demonstrated by Taylor and Merrifield in their seminal 2011 PLoS paper & it should be briefly described and referenced

Response: Thank you for this correction. We have added brief descriptions of the paper from Taylor's group in the revised manuscript (page 3, lines 44–45):

‘Asymmetric distribution revealed via total internal reflection fluorescence microscopy is an interesting characteristic of the actin cytoskeleton around CCP²².’

3. *Syndapin2 is more often called Pacsin2 and this should be stated. A study from the Traub lab showed that it participates in CME causing phenotypes but not inhibiting endocytosis levels ∅ this may well be explained by the data presented in this paper and should be noted.*

Response: Thank you for this comment. We have stated that Syndapin2 is also known as Pacsin2 in the revised manuscript (page 5, line 98). Moreover, we have carefully read the paper published by Traub's group:

Edeling, M.A. *et al.* Structural requirements for PACSIN/Syndapin operation during zebrafish embryonic notochord development. *PLoS One* **4**, e8150 (2009)

This wholesome study illustrates the structural properties of the EFC domain of Pacsin3 and the important role of Pacsin3 in the embryonic development of zebrafish. However, it provided limited information on Pacsin2 except in a few embryo injection experiments, such as in Fig. 4R. Moreover, a recent study by Lowe's group observed that the loss of Pacsin2 inhibits endocytosis and impairs pronephric tubule development in zebrafish (PMID: 35616009). Therefore, it might not be reasonable to discuss the study by Traub's group in this manuscript.

4. *Line 120 Do authors mean inhibition of pit closure, which is an easy concept to grasp, when they say promoted formation of CCP*

Response: Thank you for pointing this out. Here, 'promote the formation of CCP' implies that the number of newly formed CCPs per minute per μm^2 increased with treatment. We have revised the main text accordingly (page 4, lines 88–91):

'Treatment with CK666, an Arp2/3 complex inhibitor³⁰, and knockdown (KD) of N-WASP (Extended Data Fig. 1e), an upstream activator of Arp2/3³¹, promoted the frequency of newly formed CCPs while reducing the frequency of asymmetric and symmetric bulges to ~17% and 0%, respectively (Fig. 1b, Extended Data Figs. 1f, 1g).'

5. *Line 177 ∅ for ease and completeness times from clathrin disappearance as well as closure should be listed on line 177 ∅ changing from one metric to another is confusing*

Response: Thank you for this suggestion. We agree that this is unclear. Thus, we have unified the meaning of time zero as the disappearance of the clathrin signal in fluorescence and correlative imaging in the text and revised Fig. 2d accordingly (page 5, lines 113–115):

'When the disappearance of the clathrin signal was set as time 0, CIP4 was observed at -33.9 ± 6.50 s under CLSM, and the membrane bulge formed at -30.0 ± 8.16 s under the HS-AFM (Fig. 2d, Extended Data Fig. 2g), indicating a strong association between

CIP4 and bulge formation.’

6. Line 185 *ϕ* has asymmetric distribution of CIP4 actine etc been actually seen by Taraska lab using their elegant purple false coloured CLEM? If so, as I understand, it needs to be referenced.

Response: Thank you for your comment. We have carefully read the papers from the Taraska group, including the papers published in 2017:

Sochacki, K. A., Dickey, A. M., Strub, M. P. & Taraska, J. W. Endocytic proteins are partitioned at the edge of the clathrin lattice in mammalian cells. *Nat. Cell Biol.* **19**, 352–361 (2017).

These studies demonstrated the localisation of ~20 endocytic proteins using CLEM and revealed their specific localisation around the clathrin lattice. They did not reveal experimental results on CIP4, actin, N-WASP, and Arp2/3. However, we have cited their paper and briefly described their intriguing finding in the introduction (page 3, lines 45–48):

‘Recent progress in super-resolution techniques further revealed that many endocytic proteins possess unique localisation around the clathrin lattice²³ and that actin and some associated proteins, including the neural Wiskott–Aldrich syndrome protein (N-WASP) and actin-related proteins-2/3 (Arp2/3) complex, asymmetrically assemble near the CCP²⁴.’

7. Line 192 what is HR1 short for?

Response: Thank you for your comment. HR1 is short for ‘G protein-binding homology region 1’; we have added this information to the main text (page 6, line 129).

8. Do the disordered regions contain any characterised CME protein binding motifs eg for AP2 Clathrin Eps15 FCHO? This should be stated one way or the other.

Response: We appreciate this comment. Hitherto, no endocytic protein-binding motifs within the disordered regions of CIP4, FBP17, and Pacsin2 were known. We have added a brief description to the main text accordingly (page 6, lines 126–128):

‘The middle region between these conserved domains is less conserved and mostly disordered without any known binding motifs for other endocytic proteins (Fig. 3a, Extended Data Figs. 3d, 3e).’

9. Line 279 17% in relation to what please?

Response: Thank you for pointing this out. The frequency of the asymmetric closing in cells expressing

EGFP-Cdc42 or EGFP-Cdc42(T17N) was compared to that of the non-transfected cells. We apologise for another error here because after carefully checking the original data, the correct figure is 25% instead of 17%. We have revised the main text as follows (page 7, lines 168–169) :

‘The overexpression of a dominant-negative form of Cdc42(T17N), but not the WT, severely impaired the asymmetric bulge (to ~25%) compared to that of the non-transfected cells (Fig. 4d).’

10. Line 294, is phase separation via charged unstructured portions common? A little more explanation of how this phenomenon occurs would be very instructive

Response: We appreciate the comment and apologise for our insufficient explanation of phase separation. According to the current understanding of LLPS, it can be driven by several different types of interaction; electrostatic, hydrophobic, and cation- π . The charge block interaction is one electrostatic interaction type that strongly drives LLPS. We observed such charge blocks in the FDR of CIP4 and demonstrated that they drive LLPS (Fig. 5a). Thus, we have added more information and explanations to the main text to aid understanding (page 8, lines 191–194):

‘It has been well-established that a polymer with segregated charges (i.e., block polyampholyte) exhibits stronger liquid–liquid phase separation (LLPS) than the chain with the same number of charges randomly distributed (i.e., random polyampholyte)^{37,38}, and the pattern of charge blocks is an important determinant of LLPS properties³⁹.’

11. Line 376 Please include panel or figure explainaing how CIP4/actin etc relates to amphiphysin/SNX9 BARs and dynamin for scission ρ it is unclear.

Response: We appreciate your instruction. We await direct evidence of the functional correlation between CIP4 or CIP4-induced actin polymerisation and dynamin-dependent closing. However, we could still propose some of the possible scenarios. For instance, actin- and dynamin-driven pit-closing may represent relatively independent endocytic pathways, and either could dominate depending on the free actin abundance. However, speculating that CIP4-induced actin polymerisation cooperates with dynamin in membrane scission is also reasonable. Therefore, in the revised manuscript, we have added a new section in the Discussion section and a new panel in Fig. 6 (Fig. 6f) to briefly describe these possibilities based on the current understanding of the closing CME step. Moreover, we also briefly summarised the established role of SNX9 and Amphiphysin in dynamin-dependent scission in this section. Please refer to the main text for the detailed description (pages 13 and 14, lines 338–352):

‘The BAR/F-BAR scaffolds interplay with dynamin during the membrane scission

process⁵⁶. One classic example is the opposite role of Sorting nexin 9 (SNX9) and Amphiphysin in regulating the activity of dynamin at the late stage of endocytosis. SNX9 recruits dynamin and promotes the assembly-stimulated GTPase activity of dynamin by stabilising the dynamin-membrane interaction^{57,58} (Fig. 6f). By contrast, Amphiphysin is recruited by dynamin and inhibits the dynamin ring formation by destabilising the dynamin-membrane interaction^{59,60} (Fig. 6f). Current evidence is not yet sufficient to illustrate the functional interplay between CIP4 or CIP4-mediated actin polymerisation and dynamin. Nevertheless, we have observed that dynamin depletion²⁷ and actin inhibition did not completely abolish CME progression (Fig. 1b, Extended Data Fig. 1f), suggesting that actin- and dynamin-driven pit-closing may function as relatively independent mechanisms and either could be dominant depending on free actin abundance (Fig. 6f). However, membrane tabulation induced by the F-BAR domain could be antagonised by dynamin- and membrane-associated actin cytoskeleton⁶¹. Moreover, each dynamin helix can capture 12–16 actin filaments and align them into bundles⁶². Therefore, it is also reasonable to propose that dynamin may further potentiate the CIP4-induced actin cytoskeleton reorganisation, which promotes dynamin-dependent scission processes such as the super twist of the dynamin helix⁶³. However, these speculations should be further studied.’

12. Line 385 CALM (Merrifield/Owen) epsin(deCamilli/McMahon) as well as those stated (these need referencing) primarily use their folded domains for sensing and stabilising or driving membrane curvature & the unstructured parts provide some of the mechanical force by crowding but do not sense the membrane's curvature & they do not touch it. The difference in roles should be carefully stated and explained. Even Stachowiak lab states that both are important and work in tandem.

Response: We appreciate your correction. Disordered regions mainly support the curvature-sensing and membrane-bending activity of structured domains; however, this point should be clarified. In the revised manuscript, we have rewritten the corresponding sentences and added the two references you have suggested as follows (page 10, lines 254–259):

‘It has been proposed that unstructured amino acid chains sense the curved lipid bilayer via entropic or electrostatic mechanisms, determined by their length and net charge⁴⁰. Recently, the IDRs of membrane-bound, -anchored, and -embedded proteins, including AP180, Amphiphysin 1, FBP17, CALM, and Epsin, are gaining more research attention for supporting or enhancing the membrane-bending or curvature-sensing properties of the structured domains⁴¹⁻⁴⁵. Therefore, our findings provide a novel perspective for understanding the role of disordered proteins in CME progression.’

Technical issues that need to be addressed

13. *What spatial and temporal resolutions are possible using the techniques described & these should be stated*

Response: Thank you for your comment. We have added information related to the spatial and temporal resolutions of each imaging technique in the Method section in the revised manuscript (page 17, lines 446–447 and page 18, lines 480–481, and page 19, 487–881).

14. *Lines 273-279 This is at best confusing to this reviewer. What was delay of CIP4 assembly inhibited in comparison to wt measured from? Previously two time 0 have been used & disappearance of clathrin and complete closure. Is this a third measurement? If so a standard one should be used. A value of -12.3 for a mutant is identical to the value for wt CIP4 if it is closure (line 177). However, Fig4c suggests wt suggests -26s a value identical to that for clathrin disappearance. This needs careful explaining and/or redoing/correcting depending on what the authors actually mean*

Response: Thank you for pointing this out. We apologise for the inconsistency in defining the time '0' in different experimental systems. We also noticed that you expressed a similar concern in one of the above comments (# 5). The confusion lies in using different definitions of time '0' in fluorescence and correlative imaging. Thus, we have unified the time '0' to mean the disappearance of the clathrin signal and modified all the related contents in the revised manuscript. We believe that the time course of fluorescence and HS-AFM images can now be followed more easily.

15. *Line 280 did wt CIP4 rescue the phenotype in comparison? Please state in text. If not know it should be, please do the experiment.*

Response: We appreciate this instruction. The siRNA against CIP4 used in the study targeted the F-BAR region, which will also affect the amount of overexpressed CIP4. Hence, the original manuscript did not include the rescue experiment using overexpressed CIP4.

To respond to the reviewer's comment, we re-designed another siRNA targeting the 3'UTR region of CIP4 (siCIP4_3'UTR). siCIP4_3'UTR transfection significantly reduced the endogenous but not the overexpressed CIP4 (new Extended Data Fig. 2e). Overexpression of EGFP-fused CIP4 but not empty EGFP in Cos7 cells transfected with siCIP4_3'UTR rescued the asymmetric and symmetric bulge to a similar level in non-transfected cells (Fig. 2c), indicating that CIP4 rescued the membrane bulge formation. We have added these results in the revised manuscript (page 5, lines 104–108) and revised the figure legends accordingly:

'CIP4 KD with two siRNA species (one for the coding region, siCIP4_coding; one for the 3' untranslated region, siCIP4_3'UTR) in Cos7 cells reduced the frequency of the asymmetric bulges (from ~74% to <30%) and increased the undetermined pattern

(~70%) (Fig. 2c, Extended Data Fig. 2c–2e). Moreover, overexpression of EGFP-fused CIP4 in CIP4-KD (siCIP4_3'UTR) cells rescued the asymmetric bulge (Fig. 2c, Extended Data Fig. 2d).'

16. Line 290 and again 348 *if you are going to consider multimerization state gel filtration is not adequate. SEC MALS needs to be used to avoid Rg artefacts from extended BAR domain dimers etc*

Response: We appreciate the reviewer's instruction and agree that SEC-MALS is more appropriate for measuring protein multimerisation. We contacted some researchers who are experts in SEC-MALS or DLC (dynamic light scattering) and discussed possible experiments for HR1 multimerisation. They pointed out that HR1 is too small to be analysed using these techniques; the molecules should be larger than ~10 nm in diameter—far larger than the HR1 monomer. We can enlarge it by fusing a large tag such as GST. However, it will also result in some artefacts during the multimerisation.

Therefore, we took an alternative approach (cross-linking analysis) to demonstrate the multimerisation of HR1. Purified HR1 was treated with a crosslinker with a relatively short spacer (BS³) to crosslink the multimeric form covalently. SDS-PAGE, followed by silver staining, revealed that adding the crosslinker induced two extra bands that likely corresponded to the dimeric and tetrameric forms of HR1 according to molecular size (indicated with arrowheads in the following figure). We believe this result (new Extended Data Fig. 6e) and that of gel-filtration chromatography (Extended Data Figs. c, d) demonstrate that HR1 dimerises. The corresponding description has been added to the main text (page 8, lines 184–187):

'A similar result was obtained by treating purified HR1 with bis(sulfosuccinimidyl)suberate (BS³) crosslinkers and analysing using SDS-PAGE. Adding crosslinkers induced an extra band corresponding to the HR1 dimer, but not higher-order multimers (Extended Data Fig. 6e).'

17. Line 354 *Why was SH3 not left intact and IDR1, IDR2 and IDR1+IDR2 mutated to properly dissect out*

functions of IDRs vs SH3? I think it should be if a decent explanation cannot be given.

Response: Thank you for this question. We apologise for not clarifying the purpose of this experiment. We did not aim to differentiate the function between IDRs and SH3 but provide correlations between CIP4 assembly and the FDR phase separation. We revealed that SH3 (Δ SH3) deletion did not affect CIP4 assembly at the CME site (Figs. 3b, c). We also investigated what would happen to the CIP4 assembly if we deleted the second IDR (Δ IDR2), largely reducing the LLPS propensity of CIP4-FDR (Fig. 5d). The result revealed that further deletion of IDR2 decreased the assembly rate of WT CIP4 and CIP4(Δ SH3) (Fig. 5e), with the possibility that the CIP4 assembly at the CCP is at least partially promoted by phase separation. In the revised manuscript, we have modified some related sentences to carefully interpret the result of this experiment (page 9, lines 212–217):

‘To test whether FDR-driven LLPS is essential in CIP4 assembly at the CCP, we analysed the assembly rate of CIP4 through live-cell fluorescence imaging. SH3 deletion (CIP4(Δ SH3)) did not affect the assembly rate of CIP4 at CCP (Fig. 5f). Interestingly, further deletion of IDR2(Δ IDR2/SH3), which reduces the LLPS propensity of CIP4-FDR (Figs. 5c, 5e), reduced the assembly rate by ~70% (Fig. 5f, Extended Data Fig. 6g), implicating the role of phase separation in driving CIP4 assembly at CCP.’

18. *Line 406 and earlier if you are going to state differences in affinities quantitating pull downs is not acceptable μ BLI or SPR is needed to place numbers to be quantitative*

Response: Thank you for this comment. We agree that the pull-down assay is inappropriate for comparing the ‘affinity’ of interaction, and other methods, such as SPR, are necessary. As the reviewer suggested, we have removed the word ‘affinity’ and used ‘interaction’. This is more appropriate because we did not need to compare the ‘affinity’, but their ability to interact with Cdc42. We have revised the related sentences in the main text as follows (page 11, lines 273–275):

‘Moreover, an *in vitro* pull-down assay revealed that CIP4-FDR had a higher ability to bind to Cdc42 than does the FDR of FBP17 (Fig. 4a, Extended Data Fig. 5a); this implies that disordered regions have a role in Cdc42 interactions.’

Reviewer #3 (Remarks to the Author):

The authors utilize a combination of high-speed AFM and quantitative fluorescence to elucidate a potential mechanism by which actin polymerization can drive the closing of clathrin coated pits. Their results indicate that CIP4 may be central in regulating this process. In short, key interactions between CIP4, N-WASP, Cdc42 on the plasma membrane leads to recruitment of the Arp2/3 complex which inevitably promotes actin polymerization. This polymerization proceeds into the membrane, bending it outward asymmetrically until fission is completed. The authors arrive at this conclusion by performing live cell experiments in which they

knockdown and overexpress various F-BAR containing proteins (along with their mutants) while measuring the topography of the membrane surrounding clathrin-coated pits. In addition, they perform correlative fluorescence imaging that allows them to quantify the lifetimes and colocalization of proteins on or near the pits. Further, they were able to demonstrate that CIP4 has a propensity to form liquid condensates in vivo, that actin preferentially partitions to these condensates, and that actin polymerized within these condensates. When taken together, these results compile a rather fascinating story which details the potential molecular mechanisms associated with a highly relevant physiological process. While this mechanism is speculative (and the author's do an excellent job of acknowledging that), I believe that this work is highly innovative, utilizing advanced techniques to provide new insights to questions that have remained unanswered in the field of membrane remodeling. Therefore, I believe that this manuscript would be of broad interest to readers of Nature Communications since it provides a novel framework for investigating protein- lipid interactions. I recommend this manuscript for publication after the following comments and questions are addressed.

Comments and Questions:

1. As the authors are likely aware, the “constant curvature” and “constant area” models for clathrin-coated pit maturation are highly debated within the field of membrane remodeling. The authors should provide some discussion on how their proposed mechanism fits into or is consistent with either of these models.

Response: Thank you for raising this issue. We were also concerned about it in our previous study (Yoshida *et al.*, PLOS Biol., 2018), in which we used the same correlative imaging technique between fluorescence microscopy and HS-AFM. We observed a good correlation between an increase in the pit size (diameter and depth observed using HS-AFM) and that of the clathrin signal (fluorescence microscopy). These observations support the ‘constant curvature model’ more than the ‘constant area model’. These results and discussions have been previously published; therefore, we have avoided discussing the same issue in this manuscript. Instead, we briefly mentioned it in the introduction and cited our previous paper (page 3, lines 51–56):

‘Correlative imaging of high-speed atomic force microscopy (HS-AFM) and confocal laser-scanning microscopy (CLSM) is powerful for investigating the morphological changes of the plasma membrane during endocytosis²⁶. Our previous study using this technique has revealed a good correlation between the progress of membrane invagination and the clathrin signal accumulation supporting the hotly debated ‘constant curvature model^{27,28}’. Moreover, we observed an asymmetric closing process dependent on Arp2/3-mediated actin polymerisation²⁷.’

2. The authors created an idealized, in vitro scenario for liquid-droplet formation with the isolated FDR region of CIP4. However, the actual cytosolic environment is much more complex with a variety of proteins, all with different expression levels within the cell. How does the presence of multivalent binding partners (e.g. N- WASP, Cdc42, etc.) influence the ability of this protein to form liquid droplets? Unless I missed it, the authors do not provide any data to answer this or discuss this scenario. The authors must either provide data or discuss this scenario in the manuscript.

Response: We highly appreciate this comment and agree that the intracellular environment should be considered in an *in vitro* LLPS assay. According to the reviewer’s suggestion, we have performed a series

of multi-component LLPS assays to examine how the self-assembly of CIP4 is affected by interacting partners such as N-WASP and Cdc42.

Cdc42 (WT and constitutively active form) did not undergo LLPS or enhance the LLPS of CIP4-FDR (new Fig. 5c, 5g). These results demonstrate that the Cdc42-CIP4 interaction does not contribute to the LLPS of CIP4. N-WASP, however, exhibited strong LLPS (C_{sat} : $\sim 10 \mu\text{M}$, possibly due to a long proline-rich region flanked by basic and acidic regions; new Fig. 5c), making it challenging to assess its effect on the LLPS of CIP4 via the turbidity assay (new Fig. 5g). Therefore, we performed fluorescence microscopic observation of the droplet to examine whether both proteins co-exist in the same droplet or exclude each other. As shown in Fig. 5h, N-WASP and CIP4-FDR co-existed in the same droplet. The same result was obtained with full-length CIP4 and N-WASP (new Fig. 5h). These results indicate that both proteins have a strong propensity for LLPS and tend to co-exist in the same droplet. We proposed that this mechanism may greatly accelerate the Arp2/3 complex recruitment to the CIP4-rich locus. We have added these explanations to the main text (page 9, lines 221–229):

‘As we demonstrated, CIP4 assembly is closely associated with Cdc42 and N-WASP. We then investigated whether these two regulators or effectors potentially affected the LLPS propensity of CIP4. Cdc42 (WT and constitutively active form Q61L) did not undergo LLPS or enhance CIP4-FDR LLPS (Figs. 5c, 5g). These results demonstrate that the Cdc42-CIP4 interaction does not contribute to the LLPS of CIP4. However, N-WASP exhibited strong LLPS (C_{sat} : $\sim 10 \mu\text{M}$) (Fig. 5c). Therefore, we examined whether N-WASP and CIP4 co-exist in the same droplet or exclude each other. N-WASP and full-length CIP4 co-existed in the same droplet (Fig. 5h). The same result was obtained with CIP4-FDR and N-WASP (Figs. 5g, 5h). These results indicate that both proteins have a strong propensity for LLPS and tend to co-exist in the same droplet. We propose that this mechanism markedly accelerates the Arp2/3 complex recruitment to the CIP4-rich locus, favouring actin polymerisation.’

3. Furthermore, the authors must address whether there is any evidence that the concentrations and stoichiometry of cytosolic proteins is sufficient to induce the type of phase separation that is necessary for their proposed mechanism to be valid.

Response: Thank you for pointing this out. We understand this concern and agree that it is crucial to understand CIP4’s working model. Regarding protein solution at a constant temperature, liquid–liquid phase separation could occur when the solute (protein) concentration is above a critical (saturation) concentration. In such a dispersed system, the solute is condensed into spherical droplets. However, LLPS also occurs on cellular membranes when the local protein concentration near the membrane is below the threshold. This can be achieved when the proteins are anchored to or inserted into the membrane and assembled via lateral interaction. In such a case, the protein concentration in the cytoplasm does not have to be above the critical concentration. Indeed, many recent studies reported this type of ‘membrane-

supported' LLPS at the plasma membrane and other intracellular membrane organelles (PMID: 33532844 and 32726575). This type of 'membrane-supported' LLPS also occurs on CIP4 near the CCP for two reasons. i) Interaction with Cdc42 (Fig. 4a). Cdc42 is anchored to the plasma membrane via PIP₂; thus, a large fraction of CIP4 exists on the plasma membrane or other intracellular organelles. ii) Polymerisation of F-BAR domain. The F-BAR domain polymerises on a curved membrane; therefore, CIP4 should assemble at the CCP (and other curved membranes), forming a 'core' for LLPS. The BAR domain is insufficient for the assembly at the CCP (Fig. 3b); thus, a long FDR of CIP4 is critical in recruiting other CIP4 molecules to the core and accelerating LLPS. Therefore, the total cellular amount of CIP4 does not have to be high enough to induce LLPS on the membrane. Indeed, when we quantified the endogenous CIP4 in COS7 cells via western blot, it was ~22 nM (new Extended Data Fig. 6j)—far lower than the critical concentration obtained from the *in vitro* droplet assay (~3 μM, Fig. 5c). We believe this is essential in understanding the mechanism of asymmetry; hence, we have explained it in the Discussion section (pages 11 and 12, lines 279–299):

'CIP4 strongly self-assembled and underwent LLPS *in vitro* (Fig. 5b); nonetheless, the intracellular concentration of endogenous CIP4, estimated from the total intracellular amount (~20 nM, Extended Data Fig. 6j), was far lower than the saturation concentration (~6 μM, Fig. 5c). These results suggested that in addition to the charge block-driven LLPS of FDR, a certain mechanism which assembles CIP4 near the CCP for 'nucleation' is required to initiate LLPS. One important factor in this nucleation can be stereospecific interactions involving the structured domains (F-BAR, HR1, and SH3).

In the absence of the CCP, CIP4 binds to active Cdc42 anchored to the plasma membrane via PIP₂ through the HR1 domain (Fig. 6a). CIP4 also binds to N-WASP via the SH3 domain, allowing the interaction between N-WASP and Cdc42, and ternary complexes form among CIP4, N-WASP, and Cdc42 (Fig. 6b). However, at this stage, the local concentration of CIP4 may not be sufficient to initiate LLPS (Fig. 6b). When the ternary complex occasionally approaches the CCP, the BAR domain recognises the curvature of the pit membrane and binds to it (Fig. 6c). The BAR domain multimerises on the membrane⁴⁸; therefore, it accelerates CIP4 assembly near the CCP. In good agreement with this finding, the LLPS propensity of full-length CIP4 is higher than that of FDR (Fig. 5c), suggesting the involvement of the BAR domain in LLPS progression.

Recent studies reported ‘membrane-supported’ LLPS at the plasma membrane and other intracellular membrane organelles^{49,50}, where membrane-anchored or -inserted proteins assemble on the membrane and ‘nucleate’ for LLPS. This two-dimensional system can considerably accelerate LLPS compared to a dispersed system and initiate LLPS regardless of the bulk (cytoplasmic) protein concentration being below the threshold. We speculate that these stereospecific interactions (Cdc42-HR1, F-BAR-F-BAR, and F-BAR-membrane) contribute to CIP4 nucleation near the CCP.’

4. Are the authors implying that droplets were formed with the Δ IDR1 and Δ IDR2 mutations in Figure 5d? If so, how is that possible with the absence of alternating charge in either mutant? The authors must clarify this in the manuscript.

Response: We appreciate your comment. As the reviewer pointed out, FDR without either IDR still exhibited a low but certain LLPS propensity level under extremely high protein concentrations (Fig. 5d). One possible explanation is the multimerisation of the HR1 domain. When HR1 was deleted from FDR (Δ HR1), the LLPS propensity was reduced (Fig. 5d), suggesting that HR1 multimerisation contributes to the LLPS of FDR; however, it cannot be a major driving force. We have added these explanations in the revised Results (page 8, lines 205–278) and Discussion (page 11, lines 264–268) sections:

lines 205–207

‘Deleting one of the IDRs of CIP4-FDR did not completely abolish LLPS (Fig. 5e). Therefore, we speculated that HR1 also contributes to LLPS. HR1 deletion from FDR (Δ HR1) partly, but significantly, reduced the LLPS propensity (Figs. 5c, 5e).’

lines 264–268

‘The FDRs of CIP4 and FBP17, but not the IDR of Synd2, had high LLPS propensity, and both disordered regions were required for CIP4 phase separation (Figs. 5c, 5e). However, when either of the disordered regions was deleted (Δ IDR1 or Δ IDR2), a certain level of LLPS propensity was preserved (Fig. 5e), suggesting that HR1 dimerisation may also promote the LLPS of FDR.’

5. In Figure 4D, the “none” column seems to result in an increased percentage of asymmetric structures when compared to the “non-treated” column of Figure 1B. If I’m understanding correctly, the only difference between these two columns is that Fig. 1B contains untransfected Cos7 cells while Fig. 4D contains Cos7 cells that are overexpressing EGFP. Why is there an increase in asymmetric pits when EGFP is overexpressed?

Response: Thank you for your comment. As the reviewer pointed out, the percentage of asymmetric closing in cells overexpressing EGFP seemed to increase by 10–15% compared to that of the non-transfected cells. We re-analysed our original data from four independent experiments to evaluate whether this is significant. We observed that the asymmetric closing was not significantly increased by

EGFP overexpression compared to that in non-transfected cells ($P = 0.13$; revised Fig. 4d).

Moreover, we were reminded by this comment to improve our data presentation. Therefore, we have replaced the bar graphs in the original manuscript with heat maps to reveal the frequency of closing patterns and indicate the significance compared to the control group (revised Figs. 1b, 2c, 3d, 4d and Extended Data Fig. 6i). We have also revised the figure legends accordingly.

6. In Fig. 4D, was Cdc42 knocked down in cells? If so, explicitly say so in the figure. If not, why does the expression of EGFP-Cdc42 result in a loss of asymmetric structures and increase in symmetric structures when compared to the scenario where EGFP alone is expressed (“none” column)? Is the EGFP-Cdc42 somehow interfering with the activity of the endogenous Cdc42? The authors must address these concerns in the manuscript.

Response: We appreciate your comment. As mentioned in response to the previous comment, the percentage of asymmetric closing in cells overexpressing EGFP seemed to increase by 10–15% compared to that of the non-transfected cells. We re-analysed our original data from four independent experiments to evaluate whether this is significant. We observed that the asymmetric closing was not significantly increased by EGFP overexpression compared to that in non-transfected cells ($P = 0.13$; revised Fig. 4d).

Statistical analysis revealed that EGFP-Cdc42 overexpression did not significantly increase the frequency of asymmetric closing compared to that seen with EGFP overexpression ($P = 0.15$). The symmetric closing was also not significantly increased by EGFP-Cdc42 overexpression compared to that seen with EGFP overexpression ($P = 0.30$).

7. In Figure 6d-e, the authors’ proposed mechanism depicts CIP4 liquid droplets that protrude from the membrane surface. However, the F-BAR domain’s affinity for the membrane would make this protrusion difficult and either confine any droplet formation to the 2-dimensional membrane surface, or bend the membrane inward as described by the mechanism from reference #42 in the manuscript. Assuming that an inward invagination is not formed, the authors must explain either a) how an outward protrusion of liquid droplet is feasible or b) how a planar 2-dimensional droplet can effectively sequester and promote actin assembly.

Response: We appreciate these comments and apologise for our insufficient explanation of the mechanism of membrane protrusion. We believe that the major force pushing the membrane from the cytoplasmic side of the pit is the ‘branched actin’ and not the CIP4 droplet for the following reasons. i) Actin polymerisation inhibition by CK666 inhibited the membrane protrusion (Fig. 1b and Extended Data Fig. 1f), meaning that the major driving force of the protrusion is Arp2/3-mediated actin polymerisation. ii) Electron microscopic observations revealed that branched actin fibres near the CCP are highly anisotropic. However, Arp2/3 produces branched fibre with a fixed angle ($\sim 70^\circ$). Therefore, we believe that recruiting Arp2/3 and G-actin into the CIP4 condensate accelerates the formation of branched actin fibres. The polymerisation is anisotropic; thus, actin fibres generate radial force against the membrane, resulting in membrane protrusion (upward) and membrane fusion (lateral).

To clarify the model described above, we revised Fig 6 and the related text (page 13, lines 329–337). We believe the mechanism of membrane protrusion starting from CIP4 to Arp2/3 and actin is now clearly described in the revised manuscript:

‘We speculate that ‘branched actin’, not the CIP4 condensate, generates the mechanical force to deform the plasma membrane. CK666 inhibited actin polymerisation, which consequently inhibited the membrane protrusion (Fig. 1b, Extended Data Fig. 1f), indicating that the major driving force of the protrusion is Arp2/3-mediated actin polymerisation. Electron microscopic observations revealed that branched actin fibres near the CCP are highly anisotropic¹⁷. However, Arp2/3 produces branched fibres with a fixed angle ($\sim 70^\circ$). This finding suggests that actin fibres expand in a radial orientation and push the nearby membrane, resulting in membrane bulge (upward) and fusion (lateral) (Fig. 6e). It may also generate a downward force to push the CCP toward the cell interior, as suggested by another group⁵⁵. Further studies are required to elucidate this mechanism.’

8. It is not clear how the actin polymerization is polarized in the proposed model. What aspects of this network would actively orient the polymerization in a direction that allows it to bend the membrane and not just extend into the cytosol? The authors must address this question in the manuscript.

Response: Thank you for pointing this out. Our response to the previous comment (#7) also answers this question.

9. There is extensive use of the phrase “in vivo” in the manuscript, but live cell experiments are not in vivo The authors need to fix the wording to reflect this.

Response: We appreciate this correction. We have fixed this issue in the revised manuscript.

Reviewer #1 (Remarks to the Author):

The authors have thoroughly addressed all of my comments in the revised version of the manuscript. I believe that they have also adequately addressed the comments from the other reviewers. Therefore, I recommend this manuscript for publication without any further reservations.

Reviewer #2 (Remarks to the Author):

The authors have sufficiently addressed my comments and I consider this suitable for publication in Nat Comm

Reviewer #3 (Remarks to the Author):

The authors have thoroughly and sufficiently addressed all of my comments and concerns. Therefore, I believe that the manuscript is very much suitable for publication in its current format.